# Faithful Mobile GUI Agents with Guided Advantage Estimator

**Haowen Hu** [† 1] **Pengzhou Cheng** [† 2 1] **Zheng Wu** [1] **Lingzhong Dong** [1] **Gongshen Liu** [* 1] **Zhuosheng Zhang** [* 1]

## Abstract

Vision-language model based graphical user interface (GUI) agents have shown strong interaction capabilities. However, they often behave unfaithfully, relying on memorized shortcuts rather than grounding actions in displayed screen evidence or user instructions. To address this, we propose **Faithful-Agent**, a faithfulness-first framework that reformulates GUI interaction to prioritize evidence groundedness and internal consistency. Faithful-Agent employs a two-stage pipeline: (i) a faithfulness-oriented supervised fine-tuning stage to instill abstainment behaviors under evidence perturbations; (ii) a reinforcement fine-tuning stage that further amplifies faithfulness by introducing the guided advantage estimator (GuAE), an anchor-based and variance-adaptive advantage tempering mechanism built upon group relative policy optimization. GuAE prevents advantage collapse in low-variance rollout groups under sparse GUI rewards, and with a thought-action consistency reward, Faithful-Agent elevates the Trap success rate from thirteen point eight eight percent to eighty point two one percent relative to the baseline, while preserving robust general instruction-following performance. Our code is available at https://github.com/Dreamer777hhw/Faithful-Agent.

## 1. Introduction

Agents based on Vision–language models (VLMs) have recently demonstrated remarkable proficiency in graphical user interface (GUI) interaction and instruction following, enabling end-to-end automation across mobile, web, and desktop environments (Nguyen et al., 2025; Wang et al., 2024). Taking both visual observations and textual instruc-

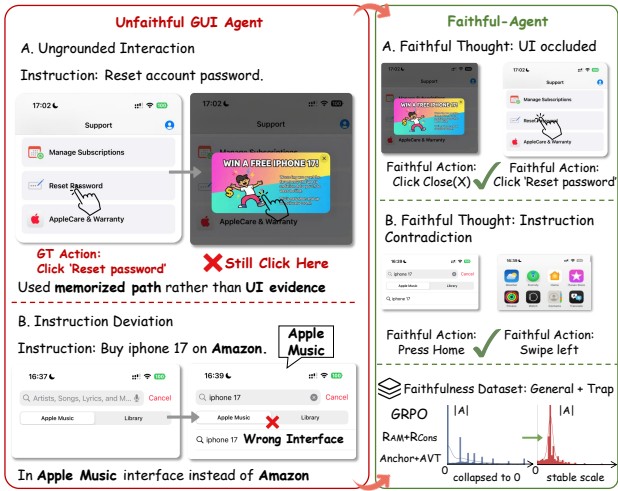

*Figure 1.* GUI agents in perturbed cases: (a) Base, where the agent takes ungrounded actions under occlusion or instruction-UI mismatch; (b) Faithful-Agent, where the agent exhibits abstainment behavior, recovers relevant states when key evidence is missing.

tions as input, these agents can either generate intermediate reasoning traces and then output executable actions (Bai et al., 2025a; Ye et al., 2025; Wang et al., 2025), or directly predict executable actions (Wu et al., 2024). Such capabilities have been significantly bolstered by recent advances in large-scale multimodal pre-training and reinforcement fine-tuning (RFT) (Yu et al., 2026; Lu et al., 2025), leading to substantial improvements in visual perception, language comprehension, and multi-step decision-making (Hong et al., 2024; Cheng et al., 2024).

However, recent research suggests that GUI agents can behave unfaithfully (Farquhar et al., 2024; Liu et al., 2025; Janiak et al., 2025; Bang et al., 2025). As illustrated in Figure 1(a), we summarize these failures into two categories. One category is **weak faithfulness to the given evidence**, where the agent does not fully condition on its inputs and instead relies on memorized shortcuts or superficial patterns (Ma et al., 2025; Cheng et al., 2025). This weakness typically manifests as (i) *Ungrounded Interaction*, where the agent proceeds without grounding decisions in the visible interface state and may take answer-like actions without verifying what is shown (Ma et al., 2025; Sun et al., 2024), and (ii) *Instruction Deviation*, where the agent deviates from

*Equal contribution [1]Shanghai Jiao Tong University, Shanghai, China [2]Shanghai University, Shanghai, China. Correspondence to: Gongshen Liu <lgshen@sjtu.edu.cn>, Zhuosheng Zhang <zhangzs@sjtu.edu.cn>.

*Proceedings of the 43rd International Conference on Machine Learning*, Seoul, South Korea. PMLR 306, 2026. Copyright 2026 by the author(s).

what the user explicitly asks for and drifts into unrelated interfaces, particularly when the instruction conflicts with the current step of the task (Zhang et al., 2025c). The other category is **internal inconsistency** (Xu et al., 2025b; Zhang et al., 2025a; Mayne et al., 2025; Dong et al., 2025), where the agent's reasoning implies one intent but its executed action follows another, leading to reasoning-action mismatch and occasionally erratic interactions.

Motivated by these observations, we reformulate GUI interaction as a *faithfulness-first problem*. Specifically, faithfulness has two core requirements, evidence groundedness and internal consistency. First, the agent's outputs should be supported by the interface evidence and the instruction (Sun et al., 2024). Second, the policy should remain internally consistent so the executed action follows what the reasoning commits to (Xu et al., 2025b). When requirements cannot be satisfied, a faithful agent should avoid speculative execution and take abstainment decisions such as terminate or pressing the back button (Feng et al., 2024; Hu et al., 2025).

Based on these principles, we introduce Faithful-Agent (Figure 1(b)), which abstains when evidence is missing or conflicting by removing occlusions, returning to relevant states, or terminating the task when necessary. To this end, we employ a two-stage training pipeline, where SFT instills abstainment behaviors while preserving general performance, and group relative policy optimization (GRPO)-based RFT further reinforces step-wise faithfulness.

During RFT, improving faithfulness is challenging under sparse GUI rewards, since faithful behavior under missing or conflicting evidence often relies on discrete enumerated abstaining actions (e.g., SYSTEM_BUTTON, TERMINATE). Thus, rollouts under the same input become difficult to differentiate within the group and often form low-variance groups. We formalize this bottleneck as **advantage collapse**: within-group normalization drives advantages toward zero in low-variance groups, weakening reward-driven updates. To address this, we introduce guided advantage estimator (GuAE), an anchor-based, variance-adaptive tempering design for base GRPO. It keeps advantages non-trivial for near-constant groups while avoiding over-scaling when variance is high, thereby preventing advantage collapse in low-variance rollout groups and enabling the policy to refine faithful actions under sparse rewards. We further add a thought-action consistency reward to better align stated intent with executed actions. Together, these designs reinforce step-wise faithfulness without sacrificing general instruction-following ability. Contributions are as follows.

(1) We define step-wise faithfulness as evidence-grounded decisions conditioned on the current interface and instruction, together with internal consistency, and specify abstainment when key evidence is missing or conflicting.

(2) We propose a two-stage training recipe for Faithful-Agent that first uses SFT to instill abstainment behaviors, then applies GRPO with GuAE, an anchor-based, variance-adaptive tempering mechanism. The second stage prevents advantage collapse and strengthens step-wise faithfulness through action-matching and thought-action consistency rewards under sparse GUI rewards.

(3) Experiments show that Faithful-Agent (Stage II) elevates the Trap SR from 13.88% to 80.21% relative to the baseline, while preserving robust general instruction-following capabilities and achieving a 71.67% overall success rate.

## 2. Related Work

### 2.1. GUI Agents and Reinforcement Fine-Tuning

Recently, VLM-based GUI agents have moved beyond grounding-centric modeling (You et al., 2024; Li et al., 2025; Gou et al., 2025) to executable end-to-end action policies (Lin et al., 2025; Zhang et al., 2025d). Several representative systems have been proposed, including large-scale cross-platform grounding (Wu et al., 2024), native end-to-end interaction from screenshots to actions (Seed, 2025; Wang et al., 2025), and scalable training for interactive learning (Ye et al., 2025), further strengthened by capable multimodal backbones such as Bai et al. (2025b) and Bai et al. (2025a). GRPO (Guo et al., 2025) enables critic-free group-relative optimization, driving RFT as a primary post-training paradigm while subsequent variants improve stability for long-horizon reasoning, e.g., DAPO (Yu et al., 2026) and GSPO (Zheng et al., 2025). Building on these optimizers, GUI-specific RFT methods further boost perception and action prediction across GUI benchmarks (Lu et al., 2026; Luo et al., 2025; Lu et al., 2025). More broadly, reasoning-oriented RFT models and objectives (He et al., 2025; Kan et al., 2025; Wei et al., 2026) offer reusable post-training ingredients for multi-step decision making that are increasingly adopted in practical GUI automation pipelines. Recent GRPO stabilization and normalization recipes, such as Durian-style difficulty-aware regrouping and DIVA-GRPO-style reward-range rescaling, are complementary to this direction. Our focus is deliberately more specific: rather than treating stabilization as a general-purpose training improvement, we study a collapse mode that arises from the action structure of faithful GUI interaction. In particular, abstainment and recovery decisions are often discrete and near-binary under outcome rewards, making low-variance rollout groups common exactly on the states where evidence-grounded behavior matters most. We take GRPO-style post-training as our backbone. While prior RL recipes largely aim at maximizing task performance, we instead optimize step-wise faithfulness of GUI agents, yielding consistently better overall results in our evaluations.

## 2.2. Faithfulness in Multimodal Models and GUI Agents

Beyond performance, *faithfulness* has attracted growing interest in VLM applications. Prior work studies faithfulness under explicit constraints and cognition-oriented perspectives (Xu et al., 2025a; Yin et al., 2025; Kargupta et al., 2025), and frames unfaithfulness as multimodal hallucination and factual misalignment, including surveys, grounding-based control, and factually augmented alignment pipelines (Favero et al., 2024; Sun et al., 2024). Reasoning traces can be unreliable indicators of faithfulness, since chain-of-thought is not necessarily explainability (Mayne et al., 2025). In GUI agents, these issues are amplified by the tight coupling of perception, reasoning, and execution, where agents may form ungrounded interface assumptions (Ma et al., 2025; Cheng et al., 2025) or exhibit words-deeds inconsistencies (Xu et al., 2025b; Dong et al., 2025). Recent work examines faithfulness under interaction hazards such as pop-up distraction and environment injection (Zhang et al., 2025b; Chen et al., 2025), proposes protocols emphasizing faithful state transitions over action matching (Zhang et al., 2024b), and reassesses whether chain-of-thought prompting improves mobile GUI agents faithfulness (Zhang et al., 2025a). Complementary directions encourage abstainment under insufficient evidence via uncertainty and clarification, explicit abstention, or reflection-based trajectory reconstruction (Feng et al., 2024; Hu et al., 2025). However, existing methods focus on assessing or eliciting faithfulness, rather than training GUI agents to be faithful to evidence. We thus bridge this gap by introducing a two-stage training pipeline that prioritizes faithfulness, enabling step-wise faithful interactions.

## 3. Preliminaries

We first introduce the problem setting in Section 3.1. Then we discuss step-wise faithfulness (Section 3.2) and advantage collapse in low-variance rollout groups (Section 3.3).

### 3.1. Problem Statement

VLM-based GUI agents interact with a mobile interface to accomplish a user instruction. At step $t$, the agent receives an input $x_t$ consisting of the instruction and the displayed interface observation (e.g., a screenshot). A policy $\pi_\theta(\cdot \mid x_t)$ samples a response $o_t = (z_t, \hat{a}_t)$, where $z_t$ is an intermediate reasoning trace and $\hat{a}_t$ is an executable action. We represent a GUI action as $\hat{a}_t = (\tau_t, \text{args}_t)$, where $\tau_t$ denotes the action type and $\text{args}_t$ are its arguments. Different backbones expose slightly different action sets; throughout this work we adopt a unified tool_call schema (Bai et al., 2025a;b). We categorize actions by how action-match is evaluated, grouping them into coordinate-based (e.g., CLICK), text-or-gesture (e.g., TYPE, SWIPE), and discrete enumerated actions (e.g., SYSTEM_BUTTON, TERMINATE), respectively.

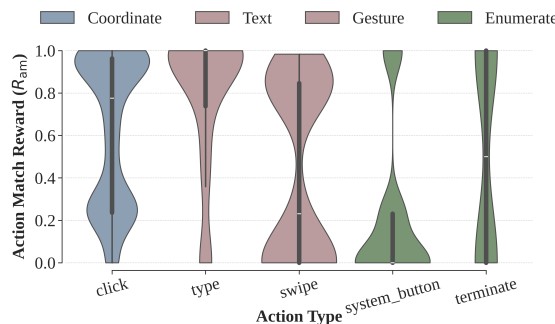

*Figure 2.* **Action-match reward distributions by action type.** Coordinate-based actions have the broadest rewards, while discrete one-of-$N$ actions are near-binary (concentrated near 0/1).

With the notations defined above, we optimize the policy $\pi_\theta$ through the following objectives. Given step inputs $x_t$, SFT trains the policy $\pi_\theta(\cdot \mid x_t)$ to imitate expert responses $o_t = (z_t, \hat{a}_t)$ from annotated trajectories. To move beyond imitation and encourage exploration among alternative step responses under the same interface evidence, we further optimize $\pi_\theta$ with outcome-supervised RFT. Specifically, for each $x_t$, we sample a rollout group of $K$ responses $\{o_{i,t}\}_{i=1}^{K} \sim \pi_\theta(\cdot \mid x_t)$, where each $o_{i,t} = (z_{i,t}, \hat{a}_{i,t})$. Each response is then assigned a reward $r_{i,t} = R(o_{i,t}, x_t) \in [0,1]$, collected in $\mathbf{r}_t = \{r_{i,t}\}_{i=1}^{K}$. We then employ GRPO to optimize the policy via within-group relative comparisons over these rewards. In this process, as our guiding principle, *faithfulness* requires $o_{i,t}$ to be grounded in $x_t$, and the action $\hat{a}_{i,t}$ to be internally consistent with $z_{i,t}$.

### 3.2. Challenges in Faithful GUI Agents

Building on the preliminaries in Section 3.1, we formally define *step-wise faithfulness* as a twofold requirement: (i) evidence groundedness, where the response $o_t$ is strictly conditioned on the interface and instruction $x_t$; (ii) internal consistency, where the executed action $\hat{a}_t$ aligns with the intent committed in the reasoning trace $z_t$. Unfaithful behaviors arise when either requirement is violated, such as acting without verifying the interface (e.g., clicking under occlusion or conflicting cues), or generating a reasoning trace $z_t$ yet executing a contradictory action. Under missing or conflicting evidence, a faithful policy should abstain via safe actions like SYSTEM_BUTTON(BACK/HOME) or TERMINATE, explicitly prioritizing backtracking or stopping until reliable cues emerge over speculative execution.

While outcome-supervised RFT with rollout groups is expected to rectify such failures, standard GRPO encounters a fundamental bottleneck in faithfulness-critical regimes. GUI interaction rewards, particularly for discrete enumerated actions (e.g., SYSTEM_BUTTON or TERMINATE), are inherently sparse or thresholded. This granularity results in an uneven reward landscape where candidate responses

within a rollout group $\{o_{i,t} = (z_{i,t}, \hat{a}_{i,t})\}_{i=1}^{K}$ often receive nearly identical scalar rewards ($r_{1,t} \approx \cdots \approx r_{K,t}$). In such low-variance scenarios, within-group preference information vanishes. We formalize this bottleneck as *advantage collapse*, which will be elaborated in the subsequent section.

### 3.3. Advantage Collapse in Faithful Agent RFT

In faithfulness-critical GUI steps, the preferred behavior is often abstainment via discrete enumerated actions. Each sampled response $o_{i,t}$ is scored by a scalar reward $r_{i,t} = R(o_{i,t}, x_t) \in [0, 1]$, and action-match uses action-type-specific rules. Reward structures vary significantly across action types. Coordinate-based actions allow for smooth shaping via distance-based rewards (Tang et al., 2026; Yuan et al., 2026). Meanwhile, text-or-gesture actions are typically graded by heuristic match functions. In contrast, discrete enumerated actions rely on thresholded criteria, which yield near-binary outcomes and create a much sparser reward landscape. For clarity, we denote the $i$-th predicted action as $\hat{a}_{i,t} = (\hat{\tau}_{i,t}, \widehat{\text{args}}_{i,t})$ and its corresponding reference as $a_t(x_t) = (\tau_t, \text{args}_t)$. For discrete enumerated actions, a minimal action-match rule is defined as:

$$r_{i,t} = \begin{cases} 1, & \hat{a}_{i,t} = a_t(x_t), \\ \rho, & \hat{\tau}_{i,t} = \tau_t, \qquad \rho \in (0,1). \\ 0, & \text{otherwise}, \end{cases} \quad (1)$$

which yields only a few discrete reward levels and thus sparse outcomes. This action-type dependence is evident in Figure 2, where coordinate-based rewards are spread out, whereas discrete one-of-$N$ decisions concentrate near $0$ and $1$. Consequently, when rewards are collected into GRPO rollout groups $\mathbf{r}_t = \{r_{i,t}\}_{i=1}^{K}$ under the same input $x_t$, responses become difficult to differentiate and within-group variation is frequently small (even all-equal), especially as the policy becomes more deterministic during training.

We formalize this effect under the standard GRPO advantage computation. GRPO computes advantages via within-group normalization. Given a rollout group reward vector $\mathbf{r}_t = \{r_{i,t}\}_{i=1}^{K}$, the standard estimator is

$$A_{i,t} = \frac{r_{i,t} - \mu(\mathbf{r}_t)}{\sigma(\mathbf{r}_t) + \varepsilon}, \qquad \mathbf{r}_t = \{r_{i,t}\}_{i=1}^{K}, \quad (2)$$

where $\mu(\mathbf{r})$ and $\sigma(\mathbf{r})$ are group mean and standard deviation.

With advantages $\{A_{i,t}\}_{i=1}^{K}$, GRPO updates $\pi_\theta$ by maximizing a KL-regularized objective against a frozen reference policy $\pi_{\text{ref}}$, a fixed snapshot of the policy at the start of RFT:

$$\mathcal{J}_{\text{GRPO}}(\theta) = \frac{1}{K} \sum_{i=1}^{K} \Big( A_{i,t} \cdot \log \pi_\theta(o_i \mid x_t) \\ - \beta \, \mathbb{D}_{\text{KL}}\big(\pi_\theta(\cdot \mid x_t) \,\|\, \pi_{\text{ref}}(\cdot \mid x_t)\big) \Big), \quad (3)$$

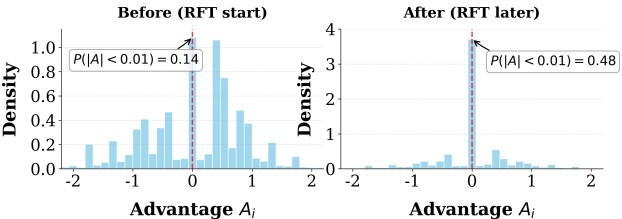

*Figure 3.* Advantage histograms under standard GRPO at the start and later stages of RFT. The distribution progressively concentrates near zero, with $P(|A| < 0.01)$ increasing from 0.14 to 0.48.

where $\beta$ controls the KL strength. Since $A_{i,t}$ directly scales the policy-gradient term, advantages that concentrate near $0$ lead to weak updates and ineffective learning signals.

A key limitation of Eq. (2) is that it becomes uninformative when rollout groups are constant. In an all-equal group where $r_i = r$ for all $i$, we have $\mu(\mathbf{r}) = r$ and therefore

$$r_i - \mu(\mathbf{r}) = 0 \quad \Rightarrow \quad A_{i,t} = 0. \quad (4)$$

Substituting Eq. (4) into Eq. (3), the policy-gradient term vanishes and the objective reduces to

$$\mathcal{J}_{\text{GRPO}}(\theta) = -\beta \, \mathbb{D}_{\text{KL}}\big(\pi_\theta(\cdot \mid x_t) \,\|\, \pi_{\text{ref}}(\cdot \mid x_t)\big). \quad (5)$$

Thus the maximizer is attained when $\pi_\theta(\cdot \mid x_t) = \pi_{\text{ref}}(\cdot \mid x_t)$, meaning that the update degenerates into pure regularization without learning from rewards. Equivalently, the gradient becomes

$$\nabla_\theta \mathcal{J}_{\text{GRPO}}(\theta) = -\beta \, \nabla_\theta \mathbb{D}_{\text{KL}}\big(\pi_\theta(\cdot \mid x_t) \,\|\, \pi_{\text{ref}}(\cdot \mid x_t)\big), \quad (6)$$

so the optimization is driven toward the reference policy.

More generally, when $\sigma(\mathbf{r})$ is so small that the denominator is dominated by $\varepsilon$, the normalization can suppress within-group variation, pushing a substantial fraction of advantages toward $0$ so that reward-driven updates become weak.

To quantify this effect, we use the near-zero mass

$$P\big(|A| < \delta\big), \quad (7)$$

which measures the fraction of rollouts that contribute negligible learning signal. Figure 3 shows that, as training proceeds, a larger fraction of advantages cluster tightly around zero, implying that updates become weak or ineffective.

Importantly, the issue is not that rewards are binary per se. Rather, as training proceeds, the policy often becomes increasingly deterministic on many states, making all-zero and all-one rollout groups common. In these groups, coarse outcome rewards provide no within-group preference signal to rank rollouts, so the update contains no reward-driven direction that favors a less-bad action among failures or a more robust action among successes. As a result, the

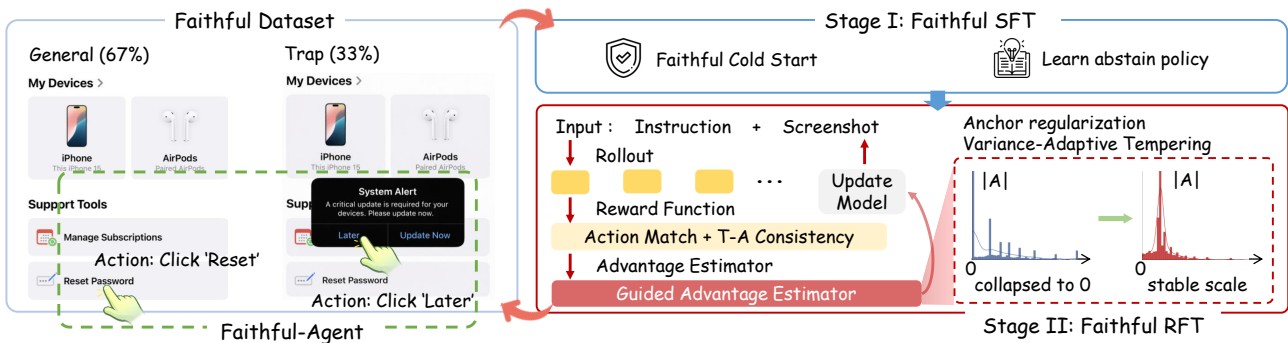

*Figure 4.* **Overview of Faithful-Agent.** We train Faithful-Agent with a two-stage pipeline: Stage I uses SFT as cold start for step-wise faithful behavior, and Stage II applies GRPO-based RFT with an action match reward and a thought-action consistency reward. During RFT, GRPO+GuAE stabilizes updates under sparse rewards and mitigates advantage collapse.

learning signal can become weakest precisely in faithfulness-critical regimes, which can become a bottleneck and limit the effectiveness of GRPO in later-stage optimization.

In Section 4, we maintain the standard GRPO objective, refine the advantage estimator under advantage collapse, and accommodate faithfulness-oriented reward shaping.

## 4. Method

To address this challenge, we propose Faithful-Agent. As illustrated in Figure 4, we first establish a faithful cold start (Stage I), followed by faithfulness-oriented RFT designed to mitigate advantage collapse (Stage II).

### 4.1. Stage I: Cold Start with Faithfulness-Oriented SFT

The goal of Stage I is to instill abstainment behaviors under missing or conflicting evidence. The SFT data supervises the agent to avoid speculative execution and instead take abstaining actions that recover evidence or exit (e.g., SYSTEM_BUTTON(BACK/HOME) or TERMINATE) when key cues are occluded or inconsistent. This cold start stabilizes RFT by reducing unfaithful exploration and providing a faithful initialization under sparse outcome rewards.

### 4.2. Stage II: GRPO with Faithfulness-Oriented Designs

In Stage II, we optimize $\pi_\theta$ via GRPO, introducing a faithfulness-first reward to enhance alignment and GuAE to mitigate advantage collapse.

#### 4.2.1. FAITHFULNESS-FIRST REWARD

For each sample, we define the reward as a convex combination of action correctness and thought-action alignment:

$$r_{i,t} = \lambda\, R_{\mathrm{AM}}(o_{i,t}) + (1-\lambda)\, R_{\mathrm{Cons}}(o_{i,t}), \quad (8)$$

where $\lambda \in [0,1]$. This ensures $r_{i,t} \in [0,1]$.

**Action-match reward.** $R_{\mathrm{AM}}$ measures whether the predicted action matches the reference action, while allowing partial credit for continuous arguments.

$$R_{\mathrm{AM}}(o_{i,t}) = \mathbb{1}[\mathrm{TYPE}(\hat{a}_{i,t}) = \mathrm{TYPE}(a_t)] \cdot \phi(\hat{a}_{i,t}, a_t), \quad (9)$$

where $\hat{a}_{i,t}$ is the parsed predicted action and $a_t$ is the ground-truth action. The indicator enforces type validity and $\phi(\hat{a}_{i,t}, a_t) \in [0,1]$ scores argument similarity (e.g., distance for coordinate actions, similarity for text actions, and exact match for discrete enumerated actions). Full per-action definitions of $\phi$ are provided in Appendix C.2.

**Thought-action consistency reward.** We encourage the executed action to match the intent expressed in the intermediate reasoning. Let $s(z_{i,t}, \hat{a}_{i,t}) \in [-1, 1]$ denote a rule-based alignment score, which we rescale to $[0, 1]$:

$$R_{\mathrm{Cons}}(o_{i,t}) = \frac{s(z_{i,t}, \hat{a}_{i,t}) + 1}{2}. \quad (10)$$

This penalizes thought-action inconsistency and complements outcome rewards by reducing speculative actions. Vacuous, generic, or underspecified intermediate reasoning receives a neutral score, avoiding misleading signals. We use this term as a bounded regularizer over explicit intent-action agreement rather than as a full semantic evaluator of reasoning quality. This design is intentionally conservative: it only checks coarse action-grounded cues that can be linked to the final executable tool call, so generic or non-checkable thoughts do not receive artificial credit. See Appendix C.3 for definition and examples of $s(\cdot, \cdot)$.

However, as the reward function is inherently sparse and near-binary (Section 3.3), a more robust estimator is necessitated to translate $r_{i,t}$ into stable advantages $A_{i,t}$.

#### 4.2.2. GUIDED ADVANTAGE ESTIMATOR (GUAE)

To address sparse-reward and low-variance rollout bottlenecks in faithfulness-critical GUI data, we propose the

Guided Advantage Estimator (GuAE) to prevent advantage collapse and preserve usable learning signals. GuAE consists of two core components: anchor regularization, which restores non-zero directional signal in fully-collapsed groups, and variance-adaptive tempering (VAT), which scales the advantage according to group dispersion to stabilize learning across diverse input steps.

**Anchor regularization.** Let $\mathbf{r}_t = \{r_{i,t}\}_{i=1}^K$ denote the rewards of a rollout group of size $K$, where $r_{i,t}$ is defined in Eq. (8).

We augment the empirical reward set with two static anchor points $\{0, 1\}$ representing the bounds of the reward space

$$\mathbf{r}_{\text{ext},t} = \{r_{1,t}, \ldots, r_{K,t}, 0, 1\}. \quad (11)$$

We then compute anchor-regularized group statistics:

$$\mu_{\text{ext},t} = \frac{1}{K+2} \sum_{r \in \mathbf{r}_{\text{ext},t}} r,$$
$$\sigma_{\text{ext},t}^2 = \frac{1}{K+2} \sum_{r \in \mathbf{r}_{\text{ext},t}} (r - \mu_{\text{ext},t})^2. \quad (12)$$

This construction yields a strictly positive normalization scale even when empirical rewards are constant. Moreover, $\sigma_{\text{ext},t}$ admits a deterministic lower bound:

$$\sigma_{\text{ext},t}^2 \geq \frac{(0 - \mu_{\text{ext},t})^2 + (1 - \mu_{\text{ext},t})^2}{K+2} \geq \frac{1}{2(K+2)},$$
$$\Rightarrow \sigma_{\text{ext},t} \geq \frac{1}{\sqrt{2(K+2)}}. \quad (13)$$

In the fully-collapsed cases $r_{i,t} \equiv c$ with $c \in \{0, 1\}$, we have $\mu_{\text{ext},t} = \frac{Kc+1}{K+2}$, and thus

$$r_{i,t} - \mu_{\text{ext},t} = c - \frac{Kc+1}{K+2} = \begin{cases} \frac{1}{K+2}, & c = 1, \\ -\frac{1}{K+2}, & c = 0. \end{cases} \quad (14)$$

In particular, $|r_{i,t} - \mu_{\text{ext},t}| = \frac{1}{K+2} \neq 0$, yielding a non-zero directional signal. While anchor regularization ensures a non-zero signal in collapsed groups, it does not account for the varying volatility across different input steps $x_t$, which we address via Variance-Adaptive Tempering. Importantly, anchor regularization does not create artificial intra-group preferences when all empirical rewards are identical. It only prevents the reward-driven update from vanishing, so the policy is no longer driven purely by the KL term in collapsed groups. The resulting signal should therefore be interpreted as preserving learnability under coarse feedback, not as ranking equally rewarded rollouts; VAT below controls the update scale across groups with different reward dispersion.

**Variance-Adaptive Tempering (VAT).** While anchors ensure signal existence, reward volatility can vary significantly across tasks, leading to signal-scale mismatch. To address this, we apply a variance-adaptive non-linear scaling.

First, we set $\sigma_0 = 1/\sqrt{12}$, the standard deviation of $\mathcal{U}(0, 1)$, serving as a non-informative reference volatility for bounded rewards (See Appendix B.4 for derivation). Then we define a normalized deviation $\delta_t$ that measures how far the group dispersion $\sigma_{\text{ext},t}$ departs from the reference baseline $\sigma_0$:

$$\delta_t = \frac{\sigma_{\text{ext},t} - \sigma_0}{\sigma_0 + \varepsilon}, \quad g_t = \text{sigmoid}\left(\tau \cdot \delta_t\right) \in (0, 1), \quad (15)$$

where $\tau > 0$ and $\varepsilon$ is a small constant. The gating factor $g_t$ interpolates a tempering exponent $p_t$ between low-variance sharpening $p_{\text{low}}$ and high-variance dampening $p_{\text{high}}$:

$$p_t = p_{\text{low}} + g_t(p_{\text{high}} - p_{\text{low}}), \quad p_{\text{low}} > 1, \ p_{\text{high}} < 1. \quad (16)$$

Tempered advantage for each sample is finally computed as:

$$A_{i,t} = \frac{r_{i,t} - \mu_{\text{ext},t}}{\sigma_{\text{ext},t}^{p_t} + \varepsilon}. \quad (17)$$

Here, $p_{\text{low}} > 1$ sharpens subtle signals in high-consensus groups, while $p_{\text{high}} < 1$ dampens noisy updates in high-variance groups.

# 5. Experiments

In this section, we evaluate Faithful-Agent by comparing it with strong open-source agents and VLM backbones, followed by analyses of our training recipes, GuAE, and reward design. We further present additional ablations and transfer results to assess robustness and generalization.

## 5.1. Implementation Details

**Experimental Setup.** We build our faithfulness-first agent based on Qwen3-VL-8B-Instruct (Bai et al., 2025a) and adopt the LLaMa-Factory (Zheng et al., 2024) framework for Stage I, and EasyR1 (Sheng et al., 2025) framework for Stage II. Our dataset is cross-scenario, consisting of correct General trajectories (General) and controlled Trap perturbations (Trap) such as occlusion, content tampering, and goal shifts. Trap examples are derived from original steps with controlled instruction and UI perturbations, and invalid or low-quality perturbations are filtered via automatic annotation and human quality control. In Trap cases, the original step is no longer justified by the current observation and thus requires evidence-conditioned abstainment. These controlled perturbations are intended as probes of deployment-relevant evidence conflicts rather than an exhaustive model of all real-world failures. They target common failure conditions such as stale visual assumptions, missing targets, or instruction-UI mismatch, where a faithful agent should first recover reliable evidence instead of continuing a memorized trajectory. We therefore complement the main split

*Table 1.* **Main results** on step-wise faithfulness on our faithfulness-oriented dataset. We compare zero-shot base GUI agent models with our Faithful-Agent (Stage I and Stage II). Best and second-best results in each column are **bolded** and underlined, respectively.

| Models | Overall | | | Trap | | General | | |
|---|---|---|---|---|---|---|---|---|
| | Type↑ | Grounding↑ | SR↑ | Type↑ | SR↑ | Type↑ | Grounding↑ | SR↑ |
| GUI-Owl-7B (Ye et al., 2025) | 59.04 | 52.27 | 29.49 | 26.00 | 12.11 | 82.49 | 52.55 | 41.82 |
| UI-TARS-1.5-7B (Seed, 2025) | 56.10 | 27.78 | 26.85 | 38.40 | 26.88 | 68.66 | 27.78 | 26.86 |
| OS-Atlas-Pro-7B (Wu et al., 2024) | 54.81 | 71.09 | 39.30 | 9.16 | 3.99 | **87.21** | 71.09 | 64.36 |
| Qwen2.5-VL-7B-Instruct (Bai et al., 2025b) | 59.29 | 48.86 | 29.00 | 30.58 | 14.77 | 79.66 | 48.86 | 39.10 |
| Qwen3-VL-8B-Instruct (Bai et al., 2025a) | 63.64 | 71.13 | 41.08 | 36.63 | 13.88 | 82.81 | 70.93 | 60.38 |
| **Faithful-Agent (Stage I)** | 80.87 | 75.05 | 64.99 | 82.27 | 71.20 | 79.87 | 74.85 | 60.59 |
| **Faithful-Agent (Stage II)** | **88.04** | **75.86** | **71.67** | **91.29** | **80.21** | 85.74 | **75.68** | **65.62** |

*Table 2.* Impact of GRPO variant and reward design in Stage II.

| Reward $R$ | GRPO variant | Trap | | General | |
|---|---|---|---|---|---|
| | | Type↑ | SR↑ | Type↑ | SR↑ |
| $R_{\text{AM}}$ | Base GRPO | 87.89 | 76.81 | 85.53 | 66.14 |
| $R_{\text{AM}} + R_{\text{Cons}}$ | Base GRPO | 89.51 | 78.29 | 85.43 | 64.36 |
| $R_{\text{AM}}$ | GRPO+GuAE | 89.36 | 78.58 | **86.27** | 66.35 |
| $R_{\text{AM}} + R_{\text{Cons}}$ | GRPO+GuAE | **91.29** | **80.21** | 85.74 | 65.62 |

*Table 3.* Impact of advantage estimator components. Improvements are reported w.r.t. Base GRPO.

| GRPO variant | Trap | | General | |
|---|---|---|---|---|
| | Type↑ | SR↑ | Type↑ | SR↑ |
| Base GRPO | 89.51 | 78.29 | 85.43 | 64.36 |
| + Anchor (only) | 90.55 (+1.04) | 79.32 (+1.03) | 83.86 (-1.57) | 63.84 (-0.52) |
| + VAT (only) | 84.34 (-5.17) | 72.67 (-5.62) | 85.22 (-0.21) | 65.41 (+1.05) |
| GRPO+GuAE | **91.29** (+1.78) | **80.21** (+1.92) | **85.74** (+0.31) | **65.62** (+1.26) |

with transfer, external distraction, and case-study analyses to test whether the learned behavior extends beyond the construction procedure itself. Details of the dataset are in Appendix A.2. Unless stated otherwise, Stage I and Stage II are trained on the same training split; the test split is strictly held out and never used in training.

**Baselines.** We compare Faithful-Agent with representative open-source GUI agents and VLM instruction backbones, including OS-Atlas-Pro-7B (Wu et al., 2024), UI-TARS-1.5-7B (Seed, 2025), GUI-Owl-7B (Ye et al., 2025), Qwen2.5-VL-7B-Instruct (Bai et al., 2025b), and Qwen3-VL-8B-Instruct (Bai et al., 2025a). All baselines use the same task prompts and observations.

**Metrics.** We evaluate Overall, General, and Trap splits using Type, Grounding, and SR following prior GUI-agent evaluations (Wu et al., 2024). Type measures action-type exact match, Grounding measures argument correctness, and SR counts a step as successful only when both the action type and arguments are correct.

### 5.2. Main Results

**Findings 1: Faithfulness improves without hurting general performance.** From Table 1, zero-shot models achieve consistently low success rates on Trap (Type-1). This suggests that faithfulness-critical states remain challenging for current GUI agents, where they tend to rely on heuristic shortcuts rather than conditioning on the current screen evidence and instruction. Our Stage I substantially improves Trap performance while keeping General stable, indicating

that faithfulness behaviors can be learned without sacrificing broad GUI ability. Stage II further strengthens faithfulness beyond imitation by preventing advantage collapse under sparse, near-binary rewards and translating meaningful learning signals into additional Trap gains, while maintaining strong General performance.

We further break down Trap into instruction perturbations and UI perturbations. The results are provided in Appendix D.11. The base model is particularly vulnerable under UI perturbations, consistent with weak interface-state grounding. Our two-stage training markedly improves robustness on UI perturbations and also yields clear gains under instruction perturbations. Moreover, results in Appendix D.1 show that increasing the Stage II budget consistently improves both Trap and General, while varying the Trap and General mix mainly adjusts the balance between higher Trap robustness and stronger General performance.

**Findings 2: GRPO+GuAE preserves learning signal and stabilizes GRPO.** In the ablation study of Table 2 with reward fixed to $R_{\text{AM}}$, GRPO+GuAE outperforms Base GRPO on both Trap and General, indicating gains beyond reward design from a more stable policy-gradient signal, while mitigating advantage collapse in low-variance groups. Figure 5 provides diagnostics. Base GRPO shows an increasing near-zero advantage ratio $P(|A| < 0.1)$ over steps, whereas GRPO+GuAE keeps it low and shifts advantage mass away from zero at a representative late stage, resulting in smoother gradient norms. In Appendix D.2 we vary VAT hyper-parameters, and results are stable across a reason-

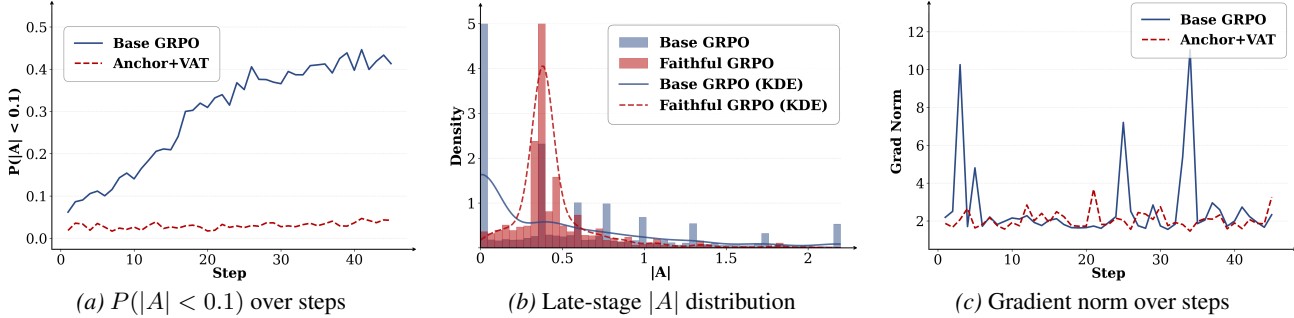

*(a) $P(|A| < 0.1)$ over steps*  *(b) Late-stage $|A|$ distribution*  *(c) Gradient norm over steps*

*Figure 5.* **GRPO+GuAE preserves advantage contrast and stabilizes Stage II training.** (a) Base GRPO yields an increasing near-zero advantage ratio, while GRPO+GuAE keeps it low. (b) At a representative late stage, GRPO+GuAE shifts mass away from zero and suppresses extreme magnitudes. (c) The resulting optimization is more stable, reflected by smoother gradient norms.

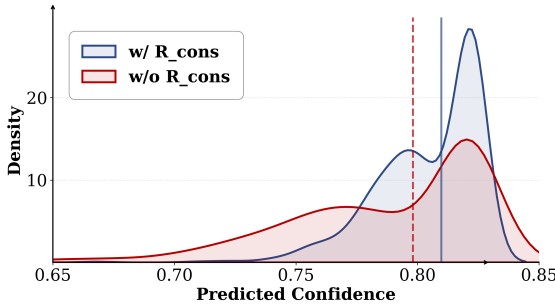

*Figure 6.* Confidence is computed from action token log probabilities and calibrated to $[0, 1]$ for visualization, where higher calibrated confidence indicates more certain action generation. Dashed lines mark the mean confidence with and without $R_{\text{Cons}}$.

*Table 4.* Training Faithful-Agent without Trap data.

| Training data | Trap | | General | |
|---|---|---|---|---|
| | Type↑ | SR↑ | Type↑ | SR↑ |
| General-only | 29.99 | 16.84 | 86.58 | 67.09 |
| Trap+General | 91.29 (+61.30) | 80.21 (+63.37) | 85.74 (-0.84) | 65.62 (-1.47) |

able range, suggesting GRPO+GuAE shows robustness to hyperparameter variations.

Table 3 isolates the two components. Anchor improves Trap by preventing advantage collapse in low-variance groups and preserving meaningful learning signal, but it can be slightly conservative on General. VAT provides adaptive gain control across groups, yet without anchors it may overly amplify noisy near-binary fluctuations, hurting Trap while slightly improving General. These trends are also reflected in Figure 7. GRPO+GuAE exhibits a clearer separation between General and Trap samples and more compact clusters than Base GRPO, with a higher silhouette score in the same reduced space. This is consistent with improved training stability under sparse, near-binary rewards.

**Findings 3: Reward design balances correctness and faithfulness.** Building on Table 1 and Table 2, we study how reward design shapes Stage II. $R_{\text{AM}}$ grounds task correctness by rewarding executed actions that match the target outcome, but under Trap perturbations it may still permit thought-action drift (e.g., intending to terminate but clicking). We therefore add $R_{\text{Cons}}$ as an auxiliary shaping signal that rewards only intent-consistent actions and penalizes contradictory thought-action pairs, while avoiding

credit for vague thoughts. Table 2 shows consistent Trap gains when adding $R_{\text{Cons}}$ to $R_{\text{AM}}$ for both Base GRPO and GRPO+GuAE, with the best overall results achieved by GRPO+GuAE under $R_{\text{AM}}+R_{\text{Cons}}$. General performance remains largely stable, indicating a reasonable trade-off: $R_{\text{AM}}$ ensures execution accuracy, while $R_{\text{Cons}}$ improves faithfulness under misleading cues.

Figure 6 further shows a more concentrated, upward-shifted confidence distribution, consistent with suppressing speculative execution. The manual audit in Appendix D.7 further confirms that $R_{\text{Cons}}$ reduces obvious rationale-action contradictions and speculative recovery rationales, while increasing UI-grounded conservative recovery. In Appendix D.8 we confirm $R_{\text{Cons}}$ works best as an auxiliary reward with moderate CLICK tolerance.

### 5.3. Analysis Experiments

**Impact of Trap data.** Table 4 shows that General-only training fails on Trap. Adding Trap data sharply improves Trap performance with minimal General impact, confirming that Trap supervision is vital for faithfulness.

**Cross-dataset generalization.** We evaluate cross-dataset transfer between AC and AITZ (Table 5). In both transfers, Stage II improves over Stage I on Trap and General, suggesting that mixing a small amount of instruction/UI-perturbed cases into general trajectories, coupled with the two-stage pipeline, is a transferable recipe for training more reliable GUI foundation models rather than dataset-specific tuning.

**Beyond step-wise evaluation.** Appendix D.5 evaluates whether step-wise gains translate beyond our main metric.

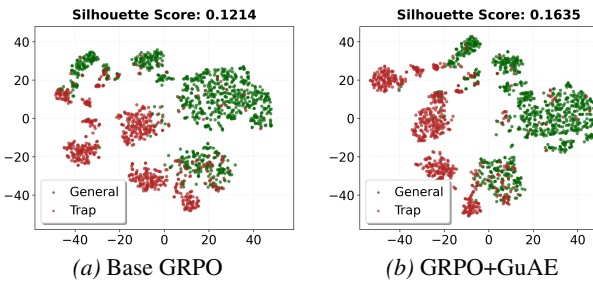

*(a)* Base GRPO      *(b)* GRPO+GuAE

*Figure 7.* Dimensionality-reduced output features under GRPO variants. We extract one output hidden-state feature per sample, project features to 2D with t-SNE, and color points by General/Trap data type. GRPO+GuAE yields clearer General/Trap separation and more compact clusters, consistent with the stability trend in Figure 5.

This matters because step-wise correctness alone does not guarantee that an agent can recover over a longer interaction after encountering missing or conflicting evidence. The consistent Stage I→Stage II gains indicate that the proposed training improves full-episode robustness and transfers to external pop-up distraction settings, providing evidence beyond our internal faithfulness-oriented split.

**Impact of RFT algorithms and advantage normalization.** Table 9 compares GRPO+GuAE with GRPO and its variants (DAPO/GSPO), plus a critic-free baseline with global advantage normalization (REINFORCE++). GRPO is strong and DAPO/GSPO are similar but slightly weaker, especially on Trap. In contrast, GRPO+GuAE achieves the best Type and SR on both splits. A more stable advantage signal improves *faithful* credit assignment: it preserves non-degenerate learning signal in low-variance groups, mitigating advantage collapse and reducing speculative behavior on Trap without hurting General performance.

**Extended Analysis.** Detailed ablation and sensitivity studies (Appendix D) reveal that (i) $\lambda$ mainly sets the Trap–General trade-off, while $\tau_{\text{click}}$ has little effect (Table 15); (ii) a moderate rollout group size $K$ is most reliable (Table 16); (iii) Stage I provides a faithful cold start by teaching abstainment (Table 17); (iv) Stage II gains concentrate on UI perturbation Traps as the base model's main failure type (Table 18); (v) RFT recalibrates abstainment, reducing under-abstain without increasing over-abstain (Table 19); (vi) GuAE outperforms curriculum-style ordering while adding negligible runtime overhead (Tables 10 and 12).

## 6. Conclusion

In this paper, we define step-wise faithfulness for GUI agents with evidence groundedness and internal consistency, and specify abstainment when key evidence is missing or conflicting. We then propose Faithful-Agent that uses SFT to teach abstainment under evidence perturbations and ap-

*Table 5.* Cross-dataset generalization (AC ↔ AITZ).

| Stage | AITZ→AC | | | | AC→AITZ | | | |
|---|---|---|---|---|---|---|---|---|
| | Trap | | General | | Trap | | General | |
| | Type↑ | SR↑ | Type↑ | SR↑ | Type↑ | SR↑ | Type↑ | SR↑ |
| Backbone | 40.55 | 16.76 | 85.66 | 61.53 | 30.91 | 7.54 | 72.60 | 53.27 |
| Stage I | 65.42 | 45.27 | 66.76 | 46.38 | 82.55 | 70.18 | 50.00 | 32.69 |
| Stage II | 67.16 | 51.74 | 68.23 | 50.67 | 85.09 | 76.73 | 58.65 | 35.10 |

plies GRPO with GuAE plus a thought-action consistency reward. Experiments show substantial faithfulness gains while maintaining strong general performance.

## Acknowledgements

This work was supported by the National Key R&D Program of China (No.2024YFC3306500), National Natural Science Foundation of China (62406188), and Natural Science Foundation of Shanghai (24ZR1440300).

## Impact Statement

Faithful-Agent aims to make mobile GUI agents more reliable by improving step-wise faithfulness under missing or conflicting interface evidence, reducing speculative actions through abstainment training and stabilized learning under sparse, near-binary rewards.

However, more capable GUI agents can also be misused to automate harmful workflows (e.g., phishing, unauthorized account actions, or privacy-invasive navigation). Since GUI observations may contain sensitive information, privacy risks may arise when storing interaction traces or sending prompts to external services. We recommend running the agent in a sandboxed environment (e.g., container or VM) with least-privilege permissions, requiring confirmation for sensitive actions, and avoiding transmission of sensitive UI content to third-party APIs.

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

## Appendix Overview

This appendix provides additional implementation details, analyses, and results that complement the main paper. Section A details the Faithful-Agent training framework and dataset construction. Section B analyzes low-variance rollout groups and advantage collapse in GUI agent GRPO. Section C specifies the faithfulness-first reward definitions and implementation details. Section D presents additional results and case studies.

## A. Details of Faithful-Agent Training Framework

In this section, we detail the two-stage training framework of Faithful-Agent and the dataset used throughout the paper. Our pipeline consists of (i) a supervised fine-tuning (SFT) stage on our *faithfulness-oriented* dataset, which establishes step-wise faithfulness together with strong general instruction-following ability, and (ii) a GRPO-based reinforcement fine-tuning (RFT) stage, which further strengthens step-wise faithfulness under interactive, sparse, and often near-binary GUI rewards. In the RFT stage, we optimize an outcome reward that combines action match with a thought-action consistency term, and adopt GuAE, an anchor regularized and variance-adaptive tempering design to mitigate advantage collapse in low-variance rollout groups. Importantly, we use the same split, test never used in training, for both SFT and RFT to keep the data distribution consistent and enable fair comparisons across training stages.

### A.1. Action Space Mapping

First of all, we parameterize the action space $\mathcal{A}$ to capture common touchscreen interactions in mobile GUI environments. Concretely, we define a finite set of structured actions:

$$\mathcal{A} = \Big\{ \text{CLICK}(x, y),\ \text{SWIPE}(x, y, x', y'),\ \text{TYPE}(t),\ \text{SYSTEM\_BUTTON}(b),\ \text{TERMINATE}(s) \Big\}. \tag{18}$$

where

- CLICK$(x, y)$ clicks the screen at $(x, y)$.

- SWIPE$(x, y, x', y')$ swipes from $(x, y)$ to $(x', y')$.

- TYPE$(t)$ inputs a text string $t \in \mathcal{V}^*$.

- SYSTEM\_BUTTON$(b)$ presses a system button $b \in \{\text{BACK}, \text{HOME}\}$.

- TERMINATE$(s)$ terminates the episode with status $s \in \{\text{SUCCESS}, \text{FAILURE}\}$.

**Coordinate convention.**   All touch coordinates use a normalized $999 \times 999$ screen space, where $(x, y)$ denotes pixels from the left and top edges. When executing on a real device with resolution $(W, H)$, we rescale coordinates by $x_{\text{pix}} = \text{round}(x/999 \cdot W)$, $y_{\text{pix}} = \text{round}(y/999 \cdot H)$. This parameterization covers both spatially grounded actions (e.g., CLICK, SWIPE) and semantic actions (e.g., TYPE), enabling a unified and reproducible action interface across mobile environments.

### A.2. Details of the Dataset

We summarize the source breakdown in Table 6. Consistent with our discussion that GUI-agent RL exhibits an inherently uneven reward landscape, we group actions into three categories (Figure 8): (i) coordinate-based actions such as CLICK, (ii) text or gesture actions such as TYPE and SWIPE, and (iii) discrete one-of-$N$ enumerated actions such as SYSTEM\_BUTTON and TERMINATE. The dataset includes general data, which contains general interaction trajectories with mixed action types, and trap data, which focuses on system-level decision-only trajectories dominated by enumerated control actions.

In trap instances, key evidence is intentionally missing or conflicting, so continuing with the original step is unsupported; the desired behavior is to abstain via safe resolutions rather than speculate with alternative UI-level actions. For UI perturbations, we include both masking-based occlusion and inpainting-based corruption to avoid shortcut learning from mask artifacts.

Overall, general data spans diverse GUI behaviors and is dominated by CLICK actions, with substantial portions of SWIPE and TYPE. In contrast, trap data consists exclusively of SYSTEM\_BUTTON actions, capturing cases where the agent must select from a small set of enumerated system controls. This composition reinforces the challenge discussed above: learning

*Table 6.* **Dataset source statistics.** Source distribution by split and data type.

| Split | Dataset | Data Type | Count |
|---|---|---|---|
| Test | AndroidControl (Li et al., 2024) | General | 746 |
| | AndroidControl | Trap | 402 |
| | AITZ | Trap | 275 |
| | AITZ (Zhang et al., 2024a) | General | 208 |
| Train | AndroidControl (Li et al., 2024) | General | 3,168 |
| | AndroidControl | Trap | 1,207 |
| | AITZ (Zhang et al., 2024a) | General | 832 |
| | AITZ | Trap | 740 |

---

**Algorithm 1** Automatic step-wise annotation

---

**Input:** UI observation and instruction $x_t$, action history $h_t$, data type $d$, ground-truth action $a_t^{\mathrm{gt}}$; if $d = \textsc{Trap}$ also include original UI $x_t^{\mathrm{orig}}$ and reference action $a_t^{\mathrm{ref}}$
**Output:** Completion $o_t$ with THOUGHT and tool-call ACTION

**if** $d = \textsc{General}$ **then**
   Prompt the model to follow $a_t^{\mathrm{gt}}$ strictly and output THOUGHT + tool-call
**else**
   Prompt the model to compare $x_t$ with $(x_t^{\mathrm{orig}}, a_t^{\mathrm{ref}})$ and output an abstainment action when evidence is missing or conflicting
**end if**
Parse the final tool-call and discard ill-formed outputs
**return** $o_t$

---

must reconcile heterogeneous reward characteristics—smoother, distance-based signals for coordinate actions versus noisy, hand-crafted rewards for "type/swipe", and naturally near-binary outcomes for one-of-$N$ enumerated decisions—which can produce an uneven reward landscape during reinforcement fine-tuning.

### A.3. Post-Processing for Automatic Dataset Construction

Our backbone follows the default Qwen3-VL generation format, which outputs natural-language THOUGHT together with a structured tool-call ACTION. We leverage this format to construct step-wise supervision. In practice, we use prompt-based automatic annotation for both the General and Trap subsets. For brevity, we omit the full prompts in the appendix and refer readers to our released code for the exact prompt templates and decoding settings.

We then apply a lightweight post-processing procedure to (i) validate the tool-call schema (action type and required arguments), (ii) normalize it into our unified action space, and (iii) filter invalid outputs (e.g., missing fields, out-of-range coordinates, or unsupported action types). For UI-perturbed trap instances, we additionally perform an amount of human filtering to remove cases with visually implausible masking/inpainting or ineffective perturbations, ensuring that the resulting supervision reflects genuine evidence-missing and conflicting scenarios rather than artifacts. Overall, we discard roughly one third of automatically constructed candidates during post-processing and human filtering, mainly due to low-quality UI perturbations or ill-formed tool-calls.

Algorithm 1 summarizes the annotation logic used in our pipeline.

### A.4. Training Environment

All experiments are conducted on $8\times$ NVIDIA H800 GPUs (80GB per GPU), with NVIDIA driver 550.144.03 and CUDA 12.4. We use PyTorch 2.8.0 and Transformers 5.0.0.dev0.

### A.5. SFT Configuration

We first perform SFT on Qwen3-VL-8B-Instruct with full-parameter fine-tuning. We train for 3 epochs using the AdamW optimizer with a peak learning rate of $1.0 \times 10^{-5}$, cosine learning-rate scheduling, and a warmup ratio of 0.1. We enable

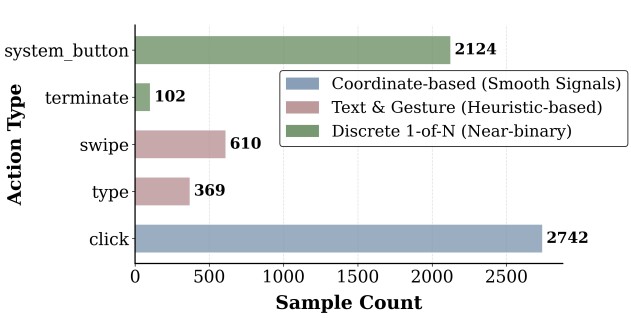

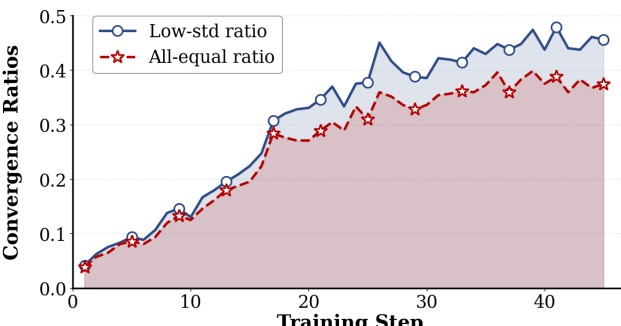

*Figure 8.* Action-type frequency in our faithfulness-oriented dataset. The most common actions are coordinate-based and discrete one-of-$N$ actions, while text and gesture actions appear less frequently.

*Figure 9.* Evolution of near-constant rollout groups during RFT. Both the low-std ratio and the all-equal ratio increase over training, indicating reduced advantage contrast in more rollout groups.

`bf16` training. The maximum sequence length is set to $8192$ tokens to accommodate long GUI observations and responses. Unless otherwise stated, all remaining SFT hyperparameters follow the default settings of LLaMA-Factory.

### A.6. RFT Configuration

We then conduct RFT using GRPO-style updates. We sample $n = 8$ rollouts per prompt with temperature $1.0$ and `top_p` $= 1.0$, and apply a KL regularization term with coefficient $1.0 \times 10^{-2}$ against a frozen reference policy to stabilize policy updates. The actor is optimized with AdamW using a learning rate of $1.0 \times 10^{-6}$ and weight decay $1.0 \times 10^{-2}$, with gradient norm clipped at $1.0$. We set the maximum prompt and response lengths to $6144$ and $2048$ tokens, respectively, and use a global update minibatch size of $192$ with micro-batches of $2$ per device for updates.

For advantage estimation, we use our modified estimator, which extends within-group normalization with anchor augmentation and variance-adaptive scaling to prevent signal collapse under near-binary reward groups. The outcome reward combines an action-correctness term with an auxiliary thought-action consistency term: we weight action match by $0.85$ and consistency by $0.15$. For coordinate-based CLICK actions, we employ a continuous distance-shaped reward with threshold $140$ pixels and decay parameter $\tau = 60$ to provide smoother learning signals, while enumerated actions naturally induce near-binary outcomes. All RFT runs are performed for $3$ epochs, and we use the same train and test split as in SFT.

## B. From Advantage Collapse to GuAE Design

This section diagnoses a core optimization pathology in GUI-agent RFT. Under sparse and near-binary outcome rewards, GRPO rollout groups frequently become low-variance or even all-equal, making within-group normalization in Eq. (2) uninformative and yielding weak learning signals. We contrast **Base GRPO** (Eq. (2)) with our **GRPO+GuAE** (Eq. (17)). Importantly, GRPO+GuAE changes only the *advantage geometry* rather than the reward, by keeping advantages non-trivial and better scaled in the collapsed regime.

### B.1. GUI Rewards Are Often Near-Binary

As discussed in Section 3.3, outcome rewards in GUI-agent RFT are often computed by matching the predicted action to a reference action or a task condition. For discrete one-of-$N$ decisions, action matching typically admits little meaningful partial credit, so the reward is effectively $0/1$, especially for abstainment and recovery actions such as SYSTEM_BUTTON and TERMINATE. For continuous actions, practical matching rules commonly rely on tolerances and string matching heuristics, which quantize the reward into only a few levels. Consequently, many faithfulness-critical steps fall into a near-binary reward regime, making low-variance rollout groups common rather than exceptional.

### B.2. Rollout Groups Collapse During Training

Consider a rollout group of size $K$ with rewards $\mathbf{r}_t = \{r_{i,t}\}_{i=1}^{K}$ and advantages computed by within-group normalization in Eq. (2). If a group collapses to a constant reward, then $r_i = \mu(\mathbf{r})$ for all $i$, so advantages vanish as shown in Eq. (4). Substituting this case into the GRPO objective in Eq. (3) yields the KL-only form in Eq. (5), meaning that the update does

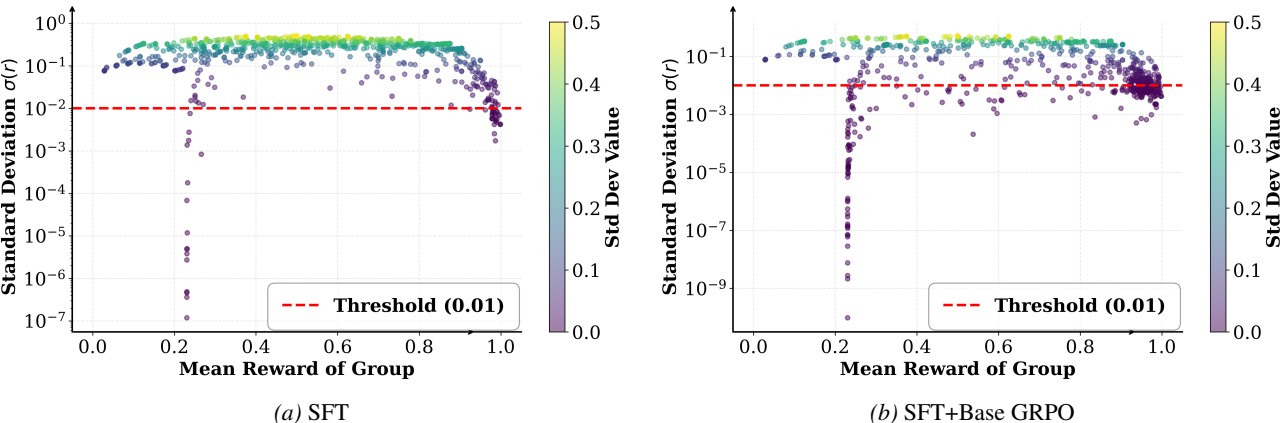

*(a)* SFT

*(b)* SFT+Base GRPO

*Figure 10.* Scatter of within-group reward variability under *vanilla* GRPO before vs. after RFT. Each point is one rollout group from a training step, plotted by its mean reward (x-axis) and within-group reward standard deviation $\sigma(r)$ (y-axis, log-scale); the dashed line marks the threshold $\sigma(r) = 0.01$.

not express any preference among rollouts within the group.

A closely related weak-signal regime occurs when the group variance is very small. When $\sigma(\mathbf{r})$ is small enough that the denominator in Eq. (2) is dominated by $\varepsilon$, many advantages shrink toward 0 and reward-driven updates become weak. We quantify this effect using the near-zero mass in Eq. (7).

Importantly, collapse is not caused by rewards being binary per se. Instead, it is promoted when outcome rewards are coarse and the policy becomes increasingly deterministic during training. Once the policy is consistently correct or consistently wrong on subsets of prompts, rollout groups naturally become all-success or all-failure, repeatedly triggering the degenerate behavior above. Empirically, Figure 9 shows that the low-std ratio and the all-equal ratio persist throughout training and tend to increase, indicating that collapsed groups are a dominant operating regime under GUI rewards.

### B.3. Diagnostics for Advantage Collapse

Collapsed groups remove within-group preference information, so GRPO receives little or no reward-driven learning signal. We monitor this phenomenon using the near-zero mass $P(|A| < \delta)$ in Eq. (7), with $\delta = 0.1$ in our experiments. A higher near-zero mass indicates that a larger fraction of rollouts contribute negligible reward-driven updates.

Our observations match this diagnostic. Figure 3 shows that advantages under Base GRPO progressively concentrate near zero during RFT, indicating systematic advantage collapse.

We further diagnose collapse at the group level using the scatter in Figure 10, where each point represents one input $x$ and its $K$ rollout rewards. Before training, groups spread over a wide range of reward means and standard deviations, indicating non-trivial within-group variation. After training, points concentrate near and below the low-variance threshold, showing that low-std groups become a dominant regime. Notably, the most severe concentration occurs near mean reward close to 1, suggesting that all-success groups are especially prone to collapsing variance once the policy becomes nearly deterministic.

### B.4. Maximum-Entropy Calibration of the Variance Threshold

Our variance-gated tempering requires a reference scale $\sigma_0$ to decide whether a rollout group should be amplified or dampened. We show that, under a minimal and axiomatic "support-only" assumption, the choice $\sigma_0 = \sqrt{1/12}$ is uniquely determined by a maximum-entropy calibration principle.

**Setup.** Let $r \in [0, 1]$ denote the (normalized) scalar reward used for group-level advantage normalization. For a rollout group, we compute an (anchor-extended) empirical standard deviation $\hat{\sigma}_{\text{ext},t}$ and define a signed, dimensionless deviation

$$d = \frac{\hat{\sigma}_{\text{ext},t} - \sigma_0}{\sigma_0 + \varepsilon}, \tag{19}$$

which is then mapped to a soft gate via a sigmoid, $g = \sigma(\tau d)$. The key question is how to choose the reference scale $\sigma_0$ without injecting unnecessary priors.

**Axioms.** We adopt the following minimal assumptions. (i) *Support-only information*: without additional prior knowledge, the only admissible information about $r$ is its support $r \in [0, 1]$. (ii) *Neutrality*: the reference distribution should be the least informative one consistent with the support constraint. (iii) *Calibration*: the gate should be centered at neutrality (e.g., $g \approx 0.5$) when the group statistics match the neutral reference.

Under these axioms, the neutral reference distribution is given by the maximum-entropy solution on $[0, 1]$.

**Maximum-entropy reference as a convex program.** Consider the set of probability density functions on $[0, 1]$, $\mathcal{P} = \{p : [0, 1] \to \mathbb{R}_{\geq 0} \mid \int_0^1 p(r)\, dr = 1\}$. We define neutrality as maximizing the (differential) entropy

$$\max_{p \in \mathcal{P}} H(p) = \max_{p \in \mathcal{P}} \left( -\int_0^1 p(r) \log p(r)\, dr \right). \tag{20}$$

Equivalently, this is the convex minimization of negative entropy,

$$\min_{p \in \mathcal{P}} \int_0^1 p(r) \log p(r)\, dr, \tag{21}$$

since the functional $\int p \log p$ is convex over $\mathcal{P}$ and the constraint set is affine.

**Solution via KKT in function space.** Introducing a Lagrange multiplier $\lambda$ for normalization yields

$$\mathcal{L}(p, \lambda) = \int_0^1 p(r) \log p(r)\, dr + \lambda \left( \int_0^1 p(r)\, dr - 1 \right). \tag{22}$$

Taking the variational derivative and setting it to zero gives

$$\frac{\delta \mathcal{L}}{\delta p}(r) = \log p(r) + 1 + \lambda = 0 \quad \Rightarrow \quad p(r) = \exp(-1 - \lambda) = \text{const.} \tag{23}$$

Enforcing $\int_0^1 p(r)\, dr = 1$ implies the unique maximizer

$$p^\star(r) = 1, \qquad r \in [0, 1], \tag{24}$$

which is the uniform distribution $\mathrm{Unif}(0, 1)$. Therefore, under support-only information, the maximum-entropy principle selects $\mathrm{Unif}(0, 1)$ as the unique bias-free reference.

**The uniquely induced reference scale.** Given a unique neutral reference distribution, its statistical scale is uniquely determined. For $r \sim \mathrm{Unif}(0, 1)$, we have

$$\mathbb{E}[r] = \frac{1}{2}, \qquad \mathrm{Var}(r) = \int_0^1 \left( r - \frac{1}{2} \right)^2 dr = \frac{1}{12}, \qquad \sigma_0 = \sqrt{\mathrm{Var}(r)} = \sqrt{\frac{1}{12}}. \tag{25}$$

This establishes $\sigma_0 = \sqrt{1/12} \approx 0.288675$ as the canonical, bias-free calibration point under the axioms above.

**Interpretation of the deviation sign.** The deviation in Eq. (19) is meaningful. When $\hat{\sigma}_{\text{ext},t} < \sigma_0$ (negative deviation), the reward mass is more concentrated than the neutral maximum-entropy reference, indicating a collapsed or highly structured regime where the effective update signal can vanish under standard normalization. When $\hat{\sigma}_{\text{ext},t} > \sigma_0$ (positive deviation), the reward values are more dispersed than the uniform baseline. Importantly, this does not imply higher entropy (uniform is already maximum entropy); instead, it indicates that probability mass shifts toward the extremes, a polarization effect that can lead to overly aggressive updates if not dampened. As a sanity check, for any $r \in [0, 1]$, Popoviciu's inequality gives

$$\mathrm{Var}(r) \leq \frac{(1 - 0)^2}{4} = \frac{1}{4}, \qquad \sigma(r) \leq \frac{1}{2}, \tag{26}$$

so $\sigma_0 = \sqrt{1/12}$ lies strictly between the collapsed regime ($\sigma \approx 0$) and the maximal polarization bound ($\sigma = 1/2$).

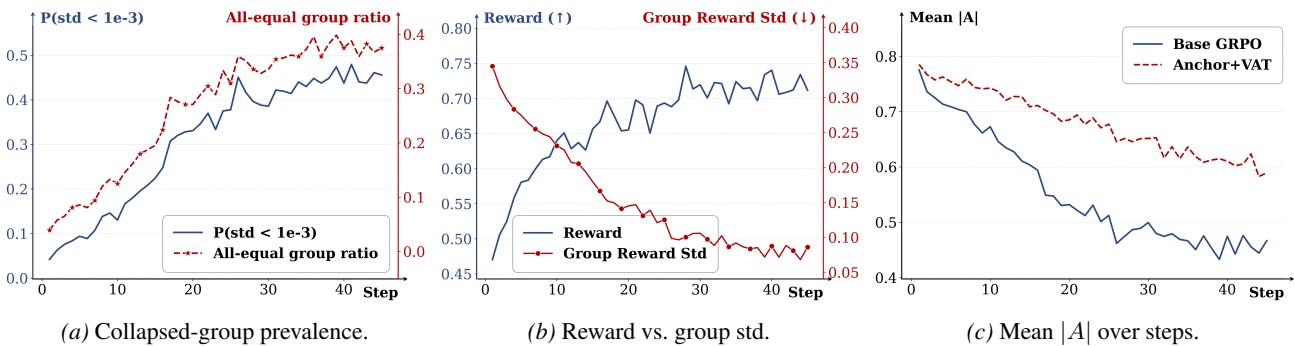

*(a)* Collapsed-group prevalence. *(b)* Reward vs. group std. *(c)* Mean $|A|$ over steps.

*Figure 11.* **Diagnosing low-variance regimes under GUI rewards.** Under *vanilla* GRPO, collapsed rollout groups become prevalent over training (a) and reward gains can coincide with shrinking within-group variance (b). Panel (c) compares *vanilla* GRPO with GRPO+GuAE, showing that GRPO+GuAE maintains larger advantage magnitudes throughout training.

**Why anchor extension is necessary.** The maximum-entropy calibration is meaningful only when the statistics are measured on a consistent support. In practice, raw rollout scores can occupy a narrow sub-interval of $[0, 1]$ (or even drift across training), making $\hat{\sigma}$ incomparable across groups. To enforce a globally consistent scale, we extend each group with anchor points at the boundaries, $r_{\text{ext},t} = \{r_i\}_{i=1}^{K} \cup \{0, 1\}$, and compute $\hat{\sigma}_{\text{ext},t}$ from $r_{\text{ext},t}$. This aligns all groups to the same support and makes the dimensionless deviation in Eq. (19) well-defined.

**Generalization to arbitrary bounded ranges.** If rewards are supported on $[a, b]$ rather than $[0, 1]$, the maximum-entropy reference remains uniform on $[a, b]$ and induces

$$\sigma_0 = \frac{b - a}{\sqrt{12}}. \tag{27}$$

Therefore, using $\sqrt{1/12}$ assumes that the reward used for gating has been normalized to $[0, 1]$, which is consistent with outcome-style rewards and with our anchor-based support alignment.

### B.5. How GRPO+GuAE Preserves Informative and Well-Scaled Advantages

GRPO+GuAE only changes the advantage computation while keeping the GRPO objective unchanged. It addresses two practical breakdowns observed in GUI-agent RFT: advantage collapse under low-variance rollout groups and signal-scale mismatch across groups with different volatility. In particular, collapsed groups can remove within-group preference information, and heterogeneous group variance can distort update scales.

**Anchor extension prevents degeneration and restores a directional signal.** We compute anchor-extended statistics in Eq. (12). The extension yields a strictly positive normalization scale with a deterministic lower bound in Eq. (13), so the denominator cannot collapse even when empirical rewards are constant. More importantly, it restores a non-zero direction in fully collapsed groups. When $r_i \equiv c$ with $c \in \{0, 1\}$, Eq. (14) gives

$$|r_i - \mu_{\text{ext},t}| = \frac{1}{K + 2} \neq 0, \tag{28}$$

so collapsed groups no longer become no-ops as in Eq. (4). This turns all-success and all-failure groups into consistent signed training signals, enabling improvement even after the policy becomes nearly deterministic on subsets of prompts.

**Variance tempering controls the gain across heterogeneous groups.** Across prompts and action types, rollout groups exhibit sharply different dispersion. A fixed mean and std normalization can mis-scale updates, where volatile groups receive overly aggressive steps while stable groups receive overly conservative steps. VAT tempers the normalization scale using the smooth gate in Eq. (15) and the interpolated exponent in Eq. (16). Since rewards are bounded, the extended standard deviation is also bounded and typically stays well below 1. For $\sigma_{\text{ext},t} \in (0, 1)$, the mapping $p \mapsto \sigma_{\text{ext},t}^p$ is decreasing. With $p_{\text{low}} > 1$ and $p_{\text{high}} < 1$, the effective denominator becomes smaller for stable groups and larger for volatile groups, which behaves as a variance-conditioned gain control and reduces scale mismatch.

**Evidence from training diagnostics.** Figure 11 characterizes the operating regime of *vanilla* GRPO. In (a), the prevalence of collapsed (near-constant) rollout groups increases steadily over training. In (b), reward improvements can occur alongside shrinking within-group variance, i.e., the regime where standard GRPO normalization tends to wash out preference information. Panel (c) then compares *vanilla* GRPO against our GRPO+GuAE, showing that GRPO+GuAE maintains larger advantage magnitudes across steps, indicating a recovered reward-driven learning signal under low-variance groups.

### B.6. Optimization Stability

GRPO+GuAE improves optimization stability by removing abrupt changes in effective normalization. Under Base GRPO, small fluctuations of $\sigma(\mathbf{r})$ around degenerate regimes can cause large relative changes in the normalization scale, leading to occasional sharp gradient spikes. Anchor extension provides a lower-bounded scale, and VAT further smooths the gain using the variance-conditioned gate, so the effective scaling varies continuously with group dispersion. As a result, optimization becomes smoother with fewer gradient-norm spikes in Figure 5c, which is consistent with stabilized advantage magnitudes in Figure 11(c) and the reduced dominance of near-threshold std groups in Figure 10b.

## C. Faithfulness-First Reward

This appendix provides the concrete definitions used to compute the two reward components in Design I: the action-match reward $R_{\mathrm{AM}}$ (Eq. 9) and the thought-action consistency reward $R_{\mathrm{Cons}}$ (Eq. 10). We emphasize that the main text focuses on the design principles; here we detail the deterministic rules used to obtain stable, reproducible supervision.

### C.1. Action Parsing and Canonicalization

Given a model completion $\mathbf{o}$, we parse it into a structured action tuple $\hat{a}$ following the unified tool-call schema used by our agent (action type plus arguments). If the completion is ill-formed (e.g., missing required keys, unparsable JSON, or empty arguments for required fields), we treat the action as invalid and set $R_{\mathrm{AM}}(\mathbf{o}) = 0$. We canonicalize both predicted and ground-truth actions into the same representation (normalized action names; normalized coordinate system; normalized text fields) before matching.

### C.2. Action-Match Reward $R_{\mathrm{AM}}$

Recall Eq. 9:
$$R_{\mathrm{AM}}(\mathbf{o}) \;=\; \mathbb{1}[\textsc{type}(\hat{a}) = \textsc{type}(a)] \cdot \phi(\hat{a}, a),$$

where $\phi(\hat{a}, a) \in [0, 1]$ is an argument similarity score. We define $\phi$ by action category to provide smoother feedback for continuous actions while preserving near-binary behavior for discrete actions.

**Coordinate actions (CLICK).** We measure Euclidean distance $d$ between predicted and ground-truth coordinates in the same normalized screen space, and use an exponential decay:

$$\phi(\hat{a}, a) \;=\; \exp(-d/\tau),$$

so small coordinate errors still receive partial credit while large deviations rapidly decay toward zero.

**Text actions (TYPE, ANSWER).** We compare the predicted string $\hat{x}$ with the target string $x$ using an edit-distance similarity:

$$\phi(\hat{a}, a) \;=\; 1 - \mathrm{EditDist}_{\mathrm{norm}}(\hat{x}, x),$$

which yields dense supervision even when text differs by minor typos or spacing.

**Swipe actions (SWIPE).** Swipe execution requires both direction and magnitude (or span) to be reasonable. We first enforce direction validity; if the direction mismatches, we set $\phi(\hat{a}, a) = 0$. Otherwise, we grant partial credit by combining a direction term with a magnitude similarity term:

$$\phi(\hat{a}, a) \;=\; \tfrac{1}{2} \;+\; \tfrac{1}{2} \cdot \psi(\hat{m}, m),$$

where $\psi(\cdot, \cdot) \in [0, 1]$ measures the similarity of swipe magnitude (e.g., relative span).

*Table 7.* RFT data sensitivity. (i) RFT data amount. (ii)Trap-to-General mixing ratio under a fixed total RFT budget of 3k samples.

| Setting | RFT | Trap | | General | |
|---|---|---|---|---|---|
| | | Type↑ | SR↑ | Type↑ | SR↑ |
| Amount | 1k | 83.75 | 72.82 | 81.76 | 63.10 |
| | 2k | 87.00 | 76.51 | 83.86 | 64.36 |
| | 4k | 89.07 | 77.55 | 84.17 | 65.30 |
| | 6k | **91.29** | **80.21** | **85.74** | **65.62** |
| Ratio | 10:1 | 74.00 | 61.00 | **86.16** | **66.25** |
| | 5:1 | 82.13 | 70.01 | 84.70 | 65.72 |
| | 2:1 | 88.21 | 77.01 | 84.03 | 64.89 |
| | 1:1 | **91.43** | **79.62** | 81.87 | 62.26 |

**Discrete enumerate actions (SYSTEM_BUTTON, TERMINATE).** For one-of-$N$ actions that represent discrete safe resolutions, the arguments are categorical and therefore matched exactly:

$$\phi(\hat{a}, a) = \mathbb{1}[\text{ARG}(\hat{a}) = \text{ARG}(a)].$$

This preserves the near-binary nature of abstainment-related decisions while keeping the overall reward bounded.

### C.3. Thought-Action Consistency Reward $R_{\text{Cons}}$

We compute a Thought-Action Consistency score $s(\text{Thought}, \hat{a}) \in [-1, 1]$ that evaluates whether the predicted action faithfully instantiates the intent stated in the intermediate reasoning (Thought). We then map it to $[0, 1]$ as in Eq. 10:

$$R_{\text{Cons}}(\mathbf{o}) = \frac{s(\text{Thought}, \hat{a}) + 1}{2}.$$

**Intent extraction.** We extract intent cues from the Thought, such as whether the plan indicates TYPING/INPUT, CLICK/TAP/PRESS, SCROLL/SWIPE, or SYSTEM_BUTTON/TERMINATE. We also extract lightweight argument cues when present (e.g., the string to be typed, swipe direction, or whether the thought implies a discrete system action).

**Alignment scoring.** The score $s(\text{Thought}, \hat{a})$ follows a rule-based rule:

- **Consistent** ($s > 0$): the predicted action type agrees with the intent cues, and any critical arguments mentioned in the thought (e.g., intended input text or swipe direction) are compatible with the action arguments.

- **Neutral** ($s \approx 0$): the thought is underspecified, ambiguous, or does not provide checkable intent cues; we neither strongly reward nor penalize.

- **Contradictory** ($s < 0$): the predicted action clearly violates the stated intent (e.g., thought says "type" but the action clicks; thought says "go back" but the action types), or required cues conflict with the action arguments.

In all cases, $s$ is clipped to $[-1, 1]$ to keep $R_{\text{Cons}}$ bounded and stable.

While our consistency reward $R_{\text{Cons}}$ relies on rule-based keyword extraction, its primary role is to act as a regularization term rather than a complex semantic evaluator. By penalizing instances where the action drifts from the explicit intent stated in the reasoning chain, $R_{\text{Cons}}$ constrains the agent's search space toward logically coherent trajectories. Our ablation study (Table 14, Table 19) and manual audit confirm that this simple yet effective signal is sufficient to suppress "hallucinated actions" without necessitating expensive LLM-based evaluators during training.

## D. More Results and Case Study

This section reports additional ablations and sensitivity analyses that do not fit in the main paper. Unless otherwise stated, all results are evaluated on the Trap and General subsets using the same metrics as in the main paper.

*Table 8.* Hyper-parameter sensitivity of VAT in GuAE. Unless otherwise specified, we use the default setting $\sigma_0$=0.2887, $\tau$=5, $p_{\text{low}}$=1.5, and $p_{\text{high}}$=0.8. Each block varies one factor while keeping the others fixed, reporting Type and SR on Trap and General.

*(a)* $(\sigma_0, \tau)$ in variance-based scaling. We vary (i) the uniform standard deviation $\sigma_0$ and (ii) the temperature $\tau$.

| Setting | Value | Trap | | General | |
|---|---|---|---|---|---|
| | | Type↑ | SR↑ | Type↑ | SR↑ |
| $\sigma_0$ | 0.2000 | 90.10 | 78.73 | 85.32 | 64.78 |
| | 0.2887 | **91.29** | **80.21** | **85.74** | **65.62** |
| | 0.3500 | 90.10 | 78.58 | 84.59 | 64.26 |
| $\tau$ | 2 | 90.10 | 79.03 | 85.22 | 65.09 |
| | 5 | **91.29** | **80.21** | **85.74** | **65.62** |
| | 10 | 89.51 | 78.58 | 84.91 | 65.51 |

*(b)* $(p_{\text{low}}, p_{\text{high}})$ in VAT exponent bounds. We vary (iii) the low-variance exponent $p_{\text{low}}$ and (iv) the high-variance exponent $p_{\text{high}}$, where $p_{\text{low}} > 1$ sharpens low-variance groups and $p_{\text{high}} < 1$ damps high-variance groups.

| Setting | Value | Trap | | General | |
|---|---|---|---|---|---|
| | | Type↑ | SR↑ | Type↑ | SR↑ |
| $p_{\text{low}}$ | 1.2 | 91.14 | 79.03 | 85.12 | 63.73 |
| | 1.5 | **91.29** | **80.21** | **85.74** | **65.62** |
| | 2.0 | 90.82 | 78.76 | 85.26 | 65.11 |
| $p_{\text{high}}$ | 0.6 | 90.99 | 78.88 | 85.85 | 65.09 |
| | 0.8 | **91.29** | **80.21** | **85.74** | **65.62** |
| | 0.9 | 91.05 | 79.03 | 85.31 | 64.89 |

*Table 9.* Impact of RFT algorithms and advantage normalization. (RFT without $R_{\text{Cons}}$)

| Optimizer | Trap | | General | |
|---|---|---|---|---|
| | Type↑ | SR↑ | Type↑ | SR↑ |
| REINFORCE++ | 81.68 | 69.57 | 83.75 | 63.21 |
| GRPO | 87.89 | 76.81 | 85.53 | 66.14 |
| DAPO | 87.59 | 76.07 | 85.64 | 64.88 |
| GSPO | 86.71 | 74.59 | 85.22 | 65.62 |
| GRPO+GuAE (Ours) | **89.36** | **78.58** | **86.27** | **66.35** |

## D.1. RFT Data Sensitivity

Table 7 studies the effect of RFT data amount and the Trap-to-General mixing ratio. Increasing the RFT budget from 1k to 6k yields consistent gains, with larger improvements on Trap, indicating that faithfulness-critical behaviors benefit more from additional RFT signal. Under a fixed total budget of 3k samples, changing the mixing ratio reveals a clear trade-off. Allocating more budget to Trap improves Trap performance but reduces General performance, while allocating too little Trap data weakens Trap robustness. The intermediate ratios provide a more balanced outcome, suggesting that Trap samples are essential but should not dominate the RFT mixture.

## D.2. Hyper-Parameter Sensitivity of Our GRPO Variant

Unless stated otherwise, we use $\sigma_0 = \frac{1}{\sqrt{12}}$, $\tau = 5$, $p_{\text{low}} = 1.5$, and $p_{\text{high}} = 0.8$. Table 8 varies key hyper-parameters in GRPO+GuAE, including the uniform standard deviation $\sigma_0$ and temperature $\tau$ in variance-based scaling, as well as the exponent bounds $(p_{\text{low}}, p_{\text{high}})$ that constrain VAT tempering. Performance remains stable across reasonable ranges of all parameters, and the default setting achieves the best or near-best results on both Trap and General. This indicates that the advantage estimator does not rely on fragile tuning and admits a robust operating region. In particular, setting $\sigma_0$ to a moderate value avoids overly aggressive scaling at small std and overly conservative scaling at large std, while varying $(p_{\text{low}}, p_{\text{high}})$ only causes minor changes, confirming that VAT is not sensitive to the exact bound choices.

## D.3. Impact of RFT Algorithms and Advantage Normalization

Table 9 compares representative group-based RFT optimizers (GRPO (Guo et al., 2025) and its variants DAPO (Yu et al., 2026) and GSPO (Zheng et al., 2025)) against REINFORCE++, a normalization-based baseline. Across both splits, standard GRPO is already strong, while DAPO and GSPO are comparable but slightly weaker overall, with more noticeable drops on Trap. In contrast, GRPO+GuAE consistently achieves the best **Type** and **SR** on both Trap and General, improving over vanilla GRPO by +1.47 Type and +1.77 SR on Trap and by +0.74 Type and +0.21 SR on General. REINFORCE++

*Table 10.* Comparison with curriculum-style ordering.

| Method | Trap | | General | |
|---|---|---|---|---|
| | **Type↑** | **SR↑** | **Type↑** | **SR↑** |
| Curriculum | 90.69 | 79.47 | 81.24 | 62.79 |
| Stage II | **91.29** | **80.21** | **85.74** | **65.62** |

*Table 11.* Robustness beyond step-wise evaluation. We report success rate (%).

| Evaluation | Backbone | Stage I | Stage II |
|---|---|---|---|
| Perturbed episodes | 41.0 | 61.1 | **67.5** |
| EnvDistraction (Ma et al., 2025) | 77.4 | 83.7 | **91.8** |

*Table 12.* Runtime overhead of GRPO+GuAE compared with vanilla GRPO.

| Metric | Base GRPO | GRPO+GuAE |
|---|---|---|
| Time per step (s) | 936.61 | 920.15 |
| Extra reward-stage time (s/step) | – | 0.10 |
| Extra advantage-stage time (s/step) | – | 0.04 |
| Reward milestone steps | Same | Same |

performs substantially worse, suggesting that applying a different normalization alone is insufficient under sparse, near-binary GUI outcomes. Overall, these results indicate that GuAE provides a more reliable policy-gradient signal than swapping the optimizer variant (DAPO/GSPO) or replacing normalization (REINFORCE++), yielding robust gains without sacrificing general performance.

### D.4. Curriculum Baseline

We compare Stage II with a curriculum-style baseline built from the same Stage I initialization and the same GRPO setup. The training samples are partitioned into three subsets by average rollout reward and then learned sequentially from easier to harder subsets. Table 10 shows that curriculum ordering improves over the starting point but remains below our full Stage II on both Trap and General, suggesting that directly addressing low-variance advantage collapse is more effective than reordering the same training data.

### D.5. Beyond Step-wise Evaluation

Table 11 evaluates whether the step-wise gains translate to broader robustness. The results show that the proposed two-stage training improves both perturbed multi-step episodes and an external pop-up distraction benchmark, supporting the main-text claim that the gains are not confined to the internal step-wise split.

### D.6. Runtime Overhead

GuAE only changes advantage computation and introduces negligible overhead relative to vanilla GRPO. As shown in Table 12, the wall-clock time is comparable to vanilla GRPO, and the extra reward and advantage computation is negligible. Both runs also reach the same reward milestones at the same training steps, indicating that GuAE improves the learning signal without adding a practical runtime burden.

### D.7. Thought-Action Consistency

Our evaluation rule is aligned with the consistency reward design in Appendix C.3, judging whether the generated thought is *consistent* with the executed action under the same criteria used by $R_{\text{Cons}}$. Table 14 shows that explicitly adding $R_{\text{Cons}}$ reliably increases the consistent portion while reducing contradictory cases, confirming that the rule is reward-aligned

*Table 13.* Manual audit of thought-action consistency on 500 randomly sampled Trap examples.

| Audit metric | w/o $R_{\mathrm{Cons}}$ | w/ $R_{\mathrm{Cons}}$ |
|---|---|---|
| Contradiction $\downarrow$ | 3.4 | **0.6** |
| Grounded recovery $\uparrow$ | 49.2 | **74.6** |
| Speculative recovery $\downarrow$ | 17.2 | **0.8** |

*Table 14.* Thought-action consistency under our rule.

| Stage | GRPO variant | Reward | Consistency | | | |
|---|---|---|---|---|---|---|
| | | | Consistent$\uparrow$ | Neutral | Contradictory$\downarrow$ | Avg. score$\uparrow$ |
| Base model | – | – | 40.10 | 0.61 | 59.29 | 0.4490 |
| Stage I | – | – | 71.00 | 1.41 | 27.59 | 0.6952 |
| Stage II | Base GRPO | – | 71.74 | 1.59 | 26.67 | 0.7039 |
| | Base GRPO | $+R_{\mathrm{Cons}}$ | **82.71** | 0.98 | **16.31** | **0.7861** |
| | GRPO+GuAE | – | 74.49 | 2.51 | 22.99 | 0.7287 |
| | GRPO+GuAE | $+R_{\mathrm{Cons}}$ | 80.44 | 2.39 | 17.17 | 0.7747 |

*Table 15.* Sensitivity to reward scaling and the click-distance threshold. We vary the reward weight $\lambda$ and the click threshold $\tau_{\mathrm{click}}$ and report Type and SR on Trap and General.

| Setting | Value | Trap | | General | |
|---|---|---|---|---|---|
| | | Type$\uparrow$ | SR$\uparrow$ | Type$\uparrow$ | SR$\uparrow$ |
| $\lambda$ | 1.00 | 89.36 | 78.58 | **86.27** | **66.35** |
| | 0.85 | 91.29 | 80.21 | 85.74 | 65.62 |
| | 0.50 | 93.65 | **82.27** | 79.98 | 60.38 |
| | 0.00 | **98.23** | 77.40 | 5.97 | 5.66 |
| $\tau_{\mathrm{click}}$ | 0 | 90.69 | 79.17 | 84.91 | 64.47 |
| | 60 | **91.29** | **80.21** | **85.74** | **65.62** |
| | 100 | 90.55 | 78.88 | 84.80 | 64.05 |
| | 140 | 90.25 | 79.03 | 84.70 | 64.78 |

and that the shaping signal is effective. Note that the rule-based consistency score is not perfectly aligned with the main task metrics; Base GRPO+RCons can be slightly higher on this auxiliary statistic. Interestingly, even without $R_{\mathrm{Cons}}$, GRPO+GuAE also improves the rule outcomes over the vanilla estimator, suggesting that thought-action consistency is not an isolated metric but is positively correlated with standard GUI optimization signals in Appendix C.2. Combining GRPO+GuAE with $R_{\mathrm{Cons}}$ maintains the improvements, yet does not consistently surpass the vanilla estimator with $R_{\mathrm{Cons}}$ under the averaged score breakdown, indicating diminishing returns and a non-trivial interaction between estimator design and consistency shaping.

In addition to the rule-based consistency score, we manually audit 500 randomly sampled Trap examples in Table 13. The audit confirms the same qualitative trend: $R_{\mathrm{Cons}}$ suppresses contradictory and speculative rationales while encouraging UI-grounded recovery rationales.

*Per-type breakdown.* For Type-0, the avg. score is 0.7162 (Base GRPO+$R_{\mathrm{Cons}}$) versus 0.6991 (GRPO+GuAE+$R_{\mathrm{Cons}}$); for Type-2, the corresponding avg. scores are 0.8846 versus 0.8812. This explains the gap mainly comes from Type-0, where behaviors are more diverse and rule consistency is not perfectly aligned with the optimization bias introduced by GRPO+GuAE. For Type-2, scores are near-saturated, so the combined method shows little additional headroom.

*Table 16.* Sensitivity to rollout group size $K$ in RFT.

| $K$ | Trap | | General | |
|---|---|---|---|---|
| | Type↑ | SR↑ | Type↑ | SR↑ |
| 4 | **91.58** | **80.21** | 83.23 | 63.10 |
| 8 | 91.29 | **80.21** | 85.74 | **65.62** |
| 16 | 87.89 | 76.66 | **86.37** | 64.99 |

*Table 17.* Impact of training stages.

| Setting | Trap | | General | |
|---|---|---|---|---|
| | Type↑ | SR↑ | Type↑ | SR↑ |
| Base (zero-shot) | 36.63 | 13.88 | 82.81 | 60.38 |
| Stage I | 82.27 | 71.20 | 79.87 | 60.59 |
| RFT only | 45.79 | 35.45 | 84.38 | **65.83** |
| Stage II | **91.29** | **80.21** | **85.74** | 65.62 |

*Table 18.* Trap subset breakdown. We decompose the Trap subset into instruction perturbations and UI perturbations, and report Type and SR on each subset.

| Models | Instruction-Trap | | UI-Trap | |
|---|---|---|---|---|
| | Type↑ | SR↑ | Type↑ | SR↑ |
| Qwen3-VL-8B-Instruct | 75.09 | 28.30 | 10.44 | 5.10 |
| **Faithful-Agent** (Stage I) | 93.58 | 67.17 | 73.79 | 71.84 |
| **Faithful-Agent** (Stage II) | **97.36** | **69.06** | **85.68** | **83.50** |

## D.8. Sensitivity to Reward Scaling and Click-Distance Threshold

Table 15 varies the reward scaling weight $\lambda$ and the click-distance threshold $\tau_{\text{click}}$. For $\lambda$, smaller values can substantially improve Trap scores but can also degrade General performance, indicating that overly strong reliance on the auxiliary reward may bias optimization toward Trap-style signals and harm generalization. The default $\lambda$ provides a better balance, while setting $\lambda = 0$ causes a collapse on General, showing that the reward term is necessary for learning meaningful behaviors. For $\tau_{\text{click}}$, the results suggest an intermediate threshold is preferred, as overly strict or overly permissive matching degrades both Trap and General due to mis-specified partial credit for coordinate actions.

## D.9. Sensitivity to Rollout Group Size

Table 16 evaluates the effect of the rollout group size $K$ in RFT. We observe that $K = 8$ yields the best overall performance, while a smaller $K$ reduces General performance and a larger $K$ hurts Trap without bringing clear benefits on General. This suggests diminishing returns from increasing within-group comparisons, and is consistent with the fact that larger groups increase the chance of low-variance outcomes under coarse rewards, reducing the fraction of informative rollouts.

## D.10. Impact of training stages

Table 17 shows a clear stage-wise pattern. Starting from the base model, *SFT provides a crucial cold start* for step-wise faithfulness: it teaches the agent to reliably *trigger abstainment* when evidence is missing or the UI state conflicts with the instruction, yielding a substantial improvement on Trap. In contrast, *RFT alone* improves overall optimization and general-task performance but offers limited gains on Trap without the correct abstainment priors. Combining them, **SFT+RFT** achieves the best results on both subsets, indicating that SFT establishes the necessary behavior primitives while RFT further strengthens and stabilizes them under sparse, near-binary rewards.

*Table 19.* Abstainment error rates. Over-abstain is computed on eligible General cases where GT is non-terminate. Under-abstain is computed on eligible Trap cases where GT requires terminate.

| Stage | GRPO variant | Reward | Abstainment errors | |
|---|---|---|---|---|
| | | | Over-abstain$\downarrow$ | Under-abstain$\downarrow$ |
| Base model | – | – | 5.24% | 57.75% |
| Stage I | – | – | 10.36% | 17.87% |
| Stage II | Base GRPO | – | 5.24% | 12.26% |
| | Base GRPO | $+R_{\mathrm{Cons}}$ | **4.78%** | 10.25% |
| | GRPO+GuAE | – | **4.78%** | 10.19% |
| | GRPO+GuAE | $+R_{\mathrm{Cons}}$ | 5.15% | **8.86%** |

## D.11. Trap Subset Breakdown

Table 18 decomposes Trap into instruction perturbations and UI perturbations. The base model performs poorly on UI-Trap, highlighting the difficulty of maintaining reliable behavior when UI evidence is missing or conflicting. Faithful-Agent trained with SFT improves both subsets substantially, indicating that the faithfulness-oriented data already teaches robust abstainment and recovery behaviors. Adding RFT further improves both subsets, with particularly substantial gains on UI-Trap, showing that our RFT stage reinforces stable decisions under visually perturbed conditions.

## D.12. Abstainment Calibration

Table 19 illustrates the evolution of decision boundaries across training stages. The base model exhibits an overconfident under-abstainment bias, characterized by severe under-abstainment on Trap cases. While SFT mitigates this, it inadvertently triggers a "conservative shift," leading to increased over-abstainment as the model overfits to termination patterns. RFT recalibrates vanilla GRPO, restoring efficiency on General while preserving Trap gains. Notably, the integration of stabilized guided advantage estimator and consistency shaping ($R_{\mathrm{Cons}}$) yields synergistic effects. This combination suppresses speculative behaviors without inducing over-caution, achieving an optimal equilibrium where the agent learns to terminate only when logically necessitated by insufficient evidence.

## D.13. Case Study

**Instruction:**
In the Calculator Unit Converter app and convert 1500 m/h into km/h units under the Speed tab.

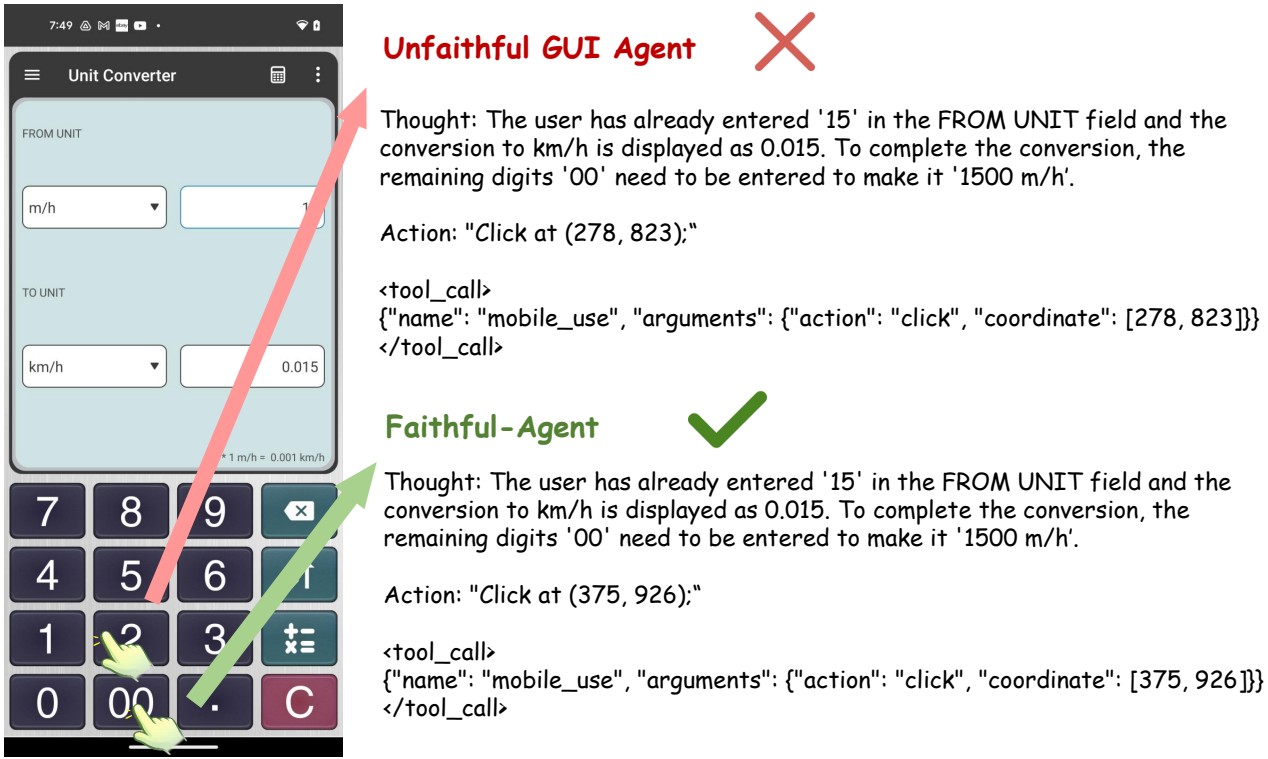

**Unfaithful GUI Agent**

Thought: The user has already entered '15' in the FROM UNIT field and the conversion to km/h is displayed as 0.015. To complete the conversion, the remaining digits '00' need to be entered to make it '1500 m/h'.

Action: "Click at (278, 823);"

<tool_call>
{"name": "mobile_use", "arguments": {"action": "click", "coordinate": [278, 823]}}
</tool_call>

**Faithful-Agent**

Thought: The user has already entered '15' in the FROM UNIT field and the conversion to km/h is displayed as 0.015. To complete the conversion, the remaining digits '00' need to be entered to make it '1500 m/h'.

Action: "Click at (375, 926);"

<tool_call>
{"name": "mobile_use", "arguments": {"action": "click", "coordinate": [375, 926]}}
</tool_call>

*Figure 12.* Thought–action mismatch in numeric entry. Both agents infer that "1500 m/h" requires appending "00", but the unfaithful agent clicks an incorrect keypad location while Faithful-Agent clicks the correct digit area to complete the input.

**Origin Instruction:**
In the The Hindu App Get the first news result from world category and first news result from Business category

**Current Instruction:**
In the Spotify App, create a new playlist titled "Morning Vibes". 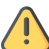

## Unfaithful GUI Agent

Current Interface

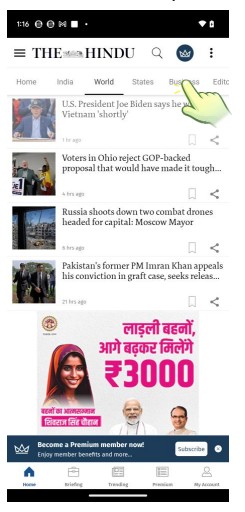

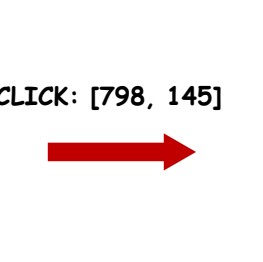

**CLICK: [798, 145]**

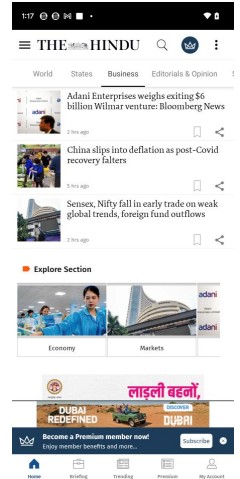

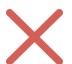

Thought: The user wants to get the first news result from both the World and Business categories. Currently, the World category is selected, and the first news item is visible. The next step is to navigate to the Business category to fetch its first news result.

Action: "Click on the 'Business' tab to switch to the Business category."

<tool_call>
{"name": "mobile_use", "arguments": {"action": "click", "coordinate": [798, 145]}}
</tool_call>",

## Faithful-Agent

Current Interface

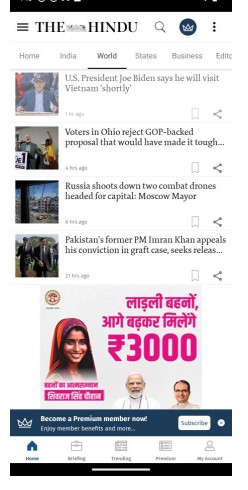

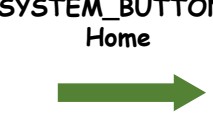

**SYSTEM_BUTTON: Home**

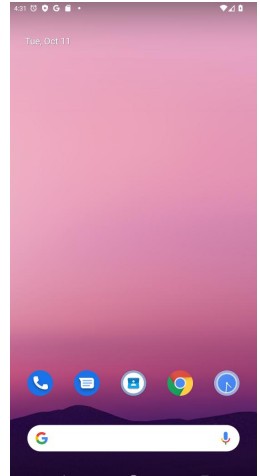

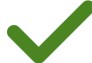

Thought: The current screen is not relevant to the Spotify app, and the task cannot be completed here.

Action: Press Home to return to the home screen and navigate to the Spotify app.

<tool_call>
{"name": "mobile_use", "arguments": {"action": "system_button", "button": "Home"}}
</tool_call>

*Figure 13.* Handling instruction drift under a fixed interface. When the current instruction switches to Spotify while the screen remains in The Hindu app, the unfaithful agent continues task execution on the irrelevant UI, whereas Faithful-Agent detects irrelevance and returns Home before re-navigation.

**Instruction:**
Open app "Microsoft Outlook" (install if not already installed) and go to login screen

## Unfaithful GUI Agent

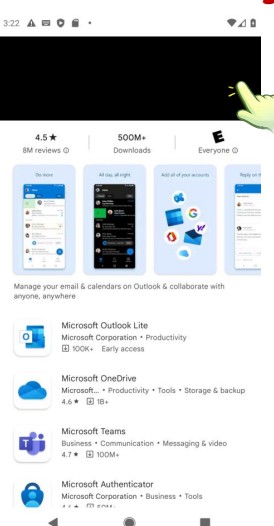

Thought: The goal is to open Microsoft Outlook and reach the login screen. Since the app is not yet installed, I need to click the "Install" button to download and install it.

Action: "Click the 'Install' button to begin installing Microsoft Outlook."

<tool_call>
{"name": "mobile_use", "arguments": {"action": "click", "coordinate": [891, 113]}}
</tool_call>

## Faithful-Agent

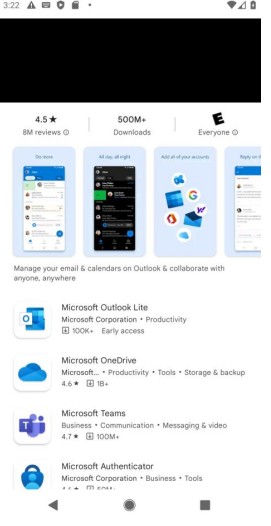

Thought: The "Install" button for Microsoft Outlook is missing, indicating the app might already be installed or the screen is altered.

Action: Press the Back button to return to the previous screen and check if the app is installed.

<tool_call>
{"name": "mobile_use", "arguments": {"action": "system_button", "button": "Back"}}
</tool_call>

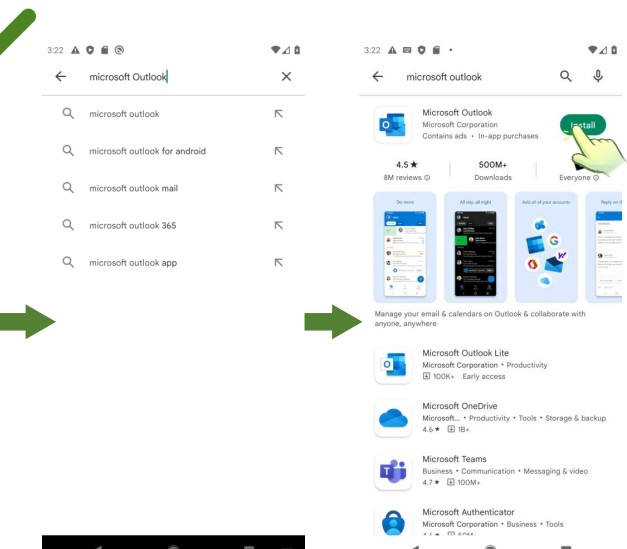

*Figure 14.* Abstaining under missing UI evidence. If the expected "Install" button is absent, the unfaithful agent still clicks the presumed location, while Faithful-Agent treats the missing element as a faithfulness warning and presses Back to verify the app state before proceeding.

