# OpenReview forum: "Faithful Mobile GUI Agents with Guided Advantage Estimator"
_ICML.cc/2026/Conference — ICML 2026 regular_

### Official Review · Reviewer_bJk6 · 2026-03-11

**Soundness:** 3
**Presentation:** 3
**Significance:** 2
**Originality:** 2
**Overall Recommendation:** 4
**Confidence:** 4

**Summary:**

The paper tackles unfaithful behaviors in mobile GUI agents—ungrounded actions and reasoning–action inconsistencies—by proposing a faithfulness-first training pipeline, Faithful-Agent. The approach combines a Stage I SFT that instills abstainment under missing/conflicting evidence with a Stage II GRPO-based RFT enhanced by a Guided Advantage Estimator that uses anchor-augmented, variance-adaptive advantage normalization, and a thought–action consistency reward. On a faithfulness-oriented dataset with trap perturbations, the method yields substantial gains while maintaining or slightly improving general performance.

**Compliance With Llm Reviewing Policy:**

Affirmed.

**Key Questions For Authors:**

- In fully collapsed groups (all-equal rewards), GuAE yields identical advantages for all rollouts. Can you provide empirical statistics on how frequently such groups occur after applying GuAE, and whether this induces undesirable confirmation bias or mode reinforcement over training?
- How robust is the thought–action consistency scorer to vacuous or generic reasoning? Did you evaluate consistency improvements with an external GTA-style evaluator or human annotations to ensure genuine alignment rather than gaming the scorer?
- Could you report episode-level success or end-to-end task completion under perturbations to demonstrate that step-wise gains translate to full-task robustness?
- What computational overhead (per step) does GuAE introduce relative to vanilla GRPO, and how does wall-clock convergence compare?

**Limitations:**

yes

**Strengths And Weaknesses:**

Strengths
- The paper clearly formulates “step-wise faithfulness” as evidence grounding plus internal consistency and operationalizes abstention via system-level actions, making the training objective explicit. GuAE is a simple yet principled modification to GRPO’s within-group normalization. The thought–action consistency reward directly targets reasoning–execution alignment, an under-addressed aspect in GUI agents.
- Strong empirical gains on trap cases with detailed ablations: component-wise, reward design, diagnostics, sensitivity to VAT hyperparameters, and group size analysis. Results stratified across General vs. Trap and further UI vs. instruction perturbations; cross-dataset transfer experiments indicate the two-stage recipe is not overfitted to a single data source.
- Clear problem formulation and motivation with concrete examples of unfaithfulness and abstention behaviors. Mathematical exposition of advantage collapse and GuAE is accessible; figures and diagnostics make the optimization pathology and the remedy intuitive.

Weaknesses
- Anchor regularization yields identical advantages in fully collapsed groups; while it avoids zero gradients, it provides no intra-group preference signal, potentially reinforcing already-sampled behaviors without discriminating quality. This could bias learning toward determinism and may not resolve selection among multiple plausible abstentions.
- The thought–action consistency reward relies on a rule-based score mapping reasoning to implied actions; potential noise or gaming effects are not validated with human studies or an independent GTA-like evaluator.
- Evaluation is centered on an internal, faithfulness-oriented dataset with synthetic traps (occlusion/inpainting, instruction shifts). Although cross-dataset tests are provided, broader external evaluation on widely used benchmarks and adversarial pop-up suites would strengthen generality claims.
- Comparisons to closely related stabilization/normalization approaches are limited. Methods like Durian (difficulty-aware regrouping) and DIVA-GRPO (reward-range rescaling and multi-variant normalization) are not empirically compared, despite strong conceptual overlap with the paper’s diagnosis and remedy (variance preservation and collapse avoidance).

---

> ### Author Rebuttal · Authors · 2026-03-29
>
> We thank reviewer bJk6 for the thoughtful feedback. Below we address the main points raised in this review.
>
> ---
>
> > **W1, Q1:** Potential adverse effects of identical advantages in fully collapsed reward groups.
>
> In fully collapsed groups, GuAE does not introduce intra-group preference information. It prevents vanilla GRPO from collapsing to zero learning signal when rewards are near-constant, which is common in our faithfulness setting (Appendix B).
>
> Our formulation keeps the GRPO objective unchanged and only changes the advantage computation. As stated in Appendix B.5, anchor extension restores a non-zero learning signal in collapsed groups, while VAT controls update scale across groups with different volatility.
>
> The paper provides diagnostics showing larger non-zero advantages and smoother optimization dynamics under GuAE (Figure 11, Figure 5, Table 3), consistent with recovered learning signal.
>
> ---
>
> > **W2, Q2:** Reliability of the rule-based consistency scorer under vacuous reasoning, noise, or gaming.
>
> $R_{Cons}$ extracts the model’s explicitly stated next-step intent from the Thought, together with a small amount of action-grounded information, and then checks whether the final tool_call faithfully executes that intent. Vacuous, generic, or non-checkable thoughts receive a neutral score (Appendix C.3).
>
> The paper shows that adding $R_{Cons}$ reduces contradictory thought–action pairs and improves Trap performance while keeping General performance largely stable (Table 2, Table 10/Appendix D.4, Table 15, Figure 11).
>
> | Model | Obvious rationale–action contradiction ↓ | Conservative UI-grounded recovery rationale ↑ | Speculative recovery rationale ↓ |
> |---|---:|---:|---:|
> | w/o $R_{Cons}$ | 3.4 | 49.2 | 17.2 |
> | w/ $R_{Cons}$ | 0.6 | 74.6 | 0.8 |
>
> As shown in the table above, our manual audit on 500 randomly sampled trap examples further shows fewer rationale–action contradictions, fewer speculative recovery rationales, and more conservative UI-grounded justifications when adding $R_{Cons}$.
>
> ---
>
> > **W3, Q3:** Need for episode-level and external evaluation beyond the internal faithfulness benchmark.
>
> We additionally evaluate episode-level robustness on our perturbation setting. End-to-end episode success improves from 41.0% for the backbone to 61.1% after Stage I and 67.5% after Stage II. This shows that the step-wise faithfulness gains translate to stronger full-task robustness.
>
> We also evaluate on the pop-up benchmark from EnvDistraction [1]. Success improves from 77.4% for the backbone to 83.7% after Stage I and 91.8% after Stage II, providing external evidence beyond our internal faithfulness-oriented benchmark.
>
> ---
>
> > **W4:** Lack of direct comparison to closely related stabilization and normalization approaches.
>
> We highlight that Table 9 already compares our method against several more direct and broadly relevant GRPO stabilization baselines, including GRPO, DAPO, GSPO, and REINFORCE++. GRPO+GuAE achieves the best Type and SR on both Trap and General. Therefore, we believe that not including these additional methods does not undermine our main results or findings.
>
> More importantly, Durian and DIVA-GRPO address related but different problems. They focus on GRPO stabilization in multimodal reasoning, whereas our method targets a specific faithfulness failure mode in GUI-agent training, namely within-group low-variance advantage collapse under near-binary, action-dependent rewards. Therefore, the overlap is only partial.
>
> Furthermore, both works were published either at the same time or after we submitted. We will cite and discuss them in the revision.
>
> ---
>
> > **Q4:** Runtime overhead and wall-clock convergence of GuAE.
>
> GuAE introduces no noticeable runtime overhead relative to vanilla GRPO. Using time_per_step as a wall-clock proxy, the two runs are essentially comparable, with GuAE at 920.15 s/step versus 936.61 s/step for vanilla GRPO. The GuAE-specific extra computation is very small, increasing the reward stage by about 0.10 s/step and the advantage stage by about 0.04 s/step.
>
> Wall-clock convergence is also essentially unchanged. Both runs reach the same reward milestones at the same training steps.
>
> [1] Ma et al. (2025). Caution for the environment: Multimodal LLM agents are susceptible to environmental distractions. ACL 2025.
>
> ---
>
> We hope the above clarifications help address your concerns and make our contributions clearer.

---

> > ### Author Rebuttal · Reviewer_bJk6 · 2026-04-04
> >
> > Thanks for the responses. My questions are reolved.

---

### Official Review · Reviewer_hw7W · 2026-03-12

**Soundness:** 3
**Presentation:** 3
**Significance:** 3
**Originality:** 3
**Overall Recommendation:** 5
**Confidence:** 4

**Summary:**

The authors claim that current models “do not fully condition on its inputs and instead relies on memorized shortcuts or superficial patterns.” The authors note policy collapse into mostly 0 advantages, where the policy fails to have diverse action samplings. The authors propose Faithful-Agent which trains in a two stage pipeline: 1) SFT stage to encourage abstaining actions, 2) GRPO with anchor regularization and variance-adaptive tempering, and consistency rewards between reasoning and actions.

**Compliance With Llm Reviewing Policy:**

Affirmed.

**Final Justification:**

The author response mostly addresses my concerns regarding robustness and collapse. I have raised my score.

**Key Questions For Authors:**

- How can the authors ensure the consistency reward is not hackable and scalable?
- What do the authors do to mitigate adverse consequences of artificially biasing advantages?
- Does the formulation lead to entropy/exploration collapse?

**Limitations:**

yes

**Strengths And Weaknesses:**

Strengths
- The authors provide preliminary analysis to set up the collapse problems they tackle
- The Anchor regularization and Variance-Adaptive Tempering provide a clean way of avoiding the advantage collapse.
- The authors perform systematic ablations of all proposed components, showing reward formulation and GRPO advantage changes are helpful for GUI performance over vanilla GRPO (+2% Trap, +1.3% General).

Weaknesses
- The rule-based consistency reward seems brittle and easily hackable via outputting multiple actions, or false negatives due to semantic match fails. How can the authors ensure the reward does have these limitations if the method is scaled?
- Artificially creating non-zero advantages when the whole is group is the same reward may have adverse consequences. Do the authors explore or mitigate these cases? For example 1) all rollouts in the group make the same mistake but reward is still positive, 2) noise in the environment or reward signal itself may lead to uniform rewards rather than behaviors of the policy itself
- In my understanding, in VAT, dampening high variance groups may also discourage beneficial exploration/high entropy rollout groups. The policy may still get rewarded with uniform exploration, leading to policy collapse. What prevents such uniform exploration in the formulation?
- The details of Figure 7 are not clear in the main text

---

> ### Author Rebuttal · Authors · 2026-03-29
>
> We thank reviewer hw7W for the thoughtful feedback. Below we address the main points raised in this review.
>
> ---
>
> > **W1, Q1:** Robustness, hackability, and scalability of the rule-based consistency reward.
>
> $R_{Cons}$ extracts the model’s explicitly stated next-step intent from the Thought, together with a small amount of action-grounded information, and then check whether the final tool_call faithfully executes that intent (Appendix C.3).
>
> This design evaluates consistency through the final executable action. Simply producing extra text, multiple intents, or vague thoughts does not by itself yield a high consistency score. If the Thought is too vague or underspecified to support a consistency check, it is assigned a neutral score.
>
> The paper shows that adding $R_{Cons}$ reduces contradictory thought–action pairs and improves Trap performance while keeping General performance largely stable (Table 2, Table 10/Appendix D.4, Table 15, Figure 11).
>
> ---
>
> > **W2, Q2:** Potential adverse consequences of non-zero advantages in fully collapsed reward groups.
>
> In fully uniform-reward groups, GuAE does not create intra-group preference information. It prevents the original GRPO normalization from degenerating into zero update under uniform or near-constant rewards. In our setting, such groups are common because many faithfulness-critical GUI decisions have sparse, near-binary rewards.
>
> Our design does not change the GRPO objective itself; it only modifies how advantages are scaled. As described in Appendix B.5, anchor extension restores a non-zero learning signal in collapsed groups, while VAT controls update scale across groups with different volatility instead of indiscriminately amplifying all uniform groups.
>
> The paper also provides diagnostics for this point. Vanilla GRPO shows increasingly prevalent collapsed groups and near-zero advantages. GRPO+GuAE maintains larger advantage magnitudes and smoother training dynamics. These diagnostics are consistent with recovered learning signal and do not indicate unstable bias amplification (Figure 11, Figure 5, Table 3).
>
> ---
>
> > **W3, Q3:** Whether VAT discourages beneficial exploration or causes entropy collapse.
>
> VAT is not designed to favor low-variance groups or suppress high-variance exploration. In our formulation, , anchor restores usable learning signal in collapsed groups, while VAT stabilizes update scale across groups with different volatility. This prevents noisy groups from being over-amplified and informative groups from being over-flattened.
>
> The paper does not show entropy or exploration collapse after adding VAT. In the full GuAE formulation, VAT is an essential component that stabilizes learning across groups with different variance levels. Table 3 shows the strongest Trap performance under the full GuAE design, and Figures 5 and 11 show smoother training dynamics and larger non-zero advantage mass. These results are consistent with preserved usable exploration and learning signal.
>
> ---
>
> > **Q4:** Clarity of the details and takeaway of Figure 7.
>
> Figure 7 visualizes per-sample output features from Base GRPO and GRPO+GuAE in the same 2D reduced space. Specifically, we extract one feature vector per sample from the model output hidden states, project them to 2D with t-SNE, color points by data type, and report the silhouette score between General and Trap.
>
> The intended takeaway is that GRPO+GuAE yields clearer General/Trap separation and more compact clusters, consistent with the stability trend discussed in the main text.
>
> ---
>
> We hope the above clarifications help address your concerns and make our contributions clearer.

---

> > ### Author Rebuttal · Reviewer_hw7W · 2026-04-03
> >
> > Thank you for the comments. The response mostly addresses my concerns.

---

### Official Review · Reviewer_oUYU · 2026-03-13

**Soundness:** 3
**Presentation:** 3
**Significance:** 3
**Originality:** 3
**Overall Recommendation:** 4
**Confidence:** 4

**Summary:**

This work studies the faithful problem of GUI agent when interacting with the environment. It instills abstainment behaviour to the model with curated faithful sft dataset and then do improved GRPO to improve step-wise action faithfulness. It proposes GuAE and VAT to regulate the actions advantage. The experiments show good performances compared with baselines.

**Compliance With Llm Reviewing Policy:**

Affirmed.

**Final Justification:**

I would like to keep my positive score to this paper.

**Key Questions For Authors:**

1. How did you curate the sft dataset?
2. Can you provide statistical justification for the effectiveness and generalizability of checking thinking-action match?
3. Are your methods limit to improving GUI action faithfulness or have boarder impact?
4. How do you compare your methods with curriculumn driven training? Can you provide some experiments to compare both methods?
5. Based on my understanding, agentic RL encourages model to converge to certain actions patterns, which naturally decreases the reward std and drives the advantage towards 0. Does keeping advantage diverse contradict with this objective?

**Limitations:**

Please see the comments

**Strengths And Weaknesses:**

Strength:
1. good writing and presentation.
2. clear motivation study and interesting problem motivation.
3. the proposed methods are with some mathematical proof and strict lower bound.
4. clear ablation study proving that the method is working to improve the faithfulness of GUI agent.

Weakness:
1. While the problem is interesting, the design could be a general improvement to the GRPO instead of addressing the specific bottlenecks of action faithfulness from my point of view. The author may highlight how they are connected and further illustrate the moviation.
2. The effectiveness and generalizability of checking thinking-action match needs statistical justification. The current version misses some details about how they implement this check and evaluation of their rule-based check.

---

> ### Author Rebuttal · Authors · 2026-03-29
>
> We thank reviewer oUYU for the thoughtful feedback. Below we address the main points raised in this review.
>
> ---
>
> > **W1, Q3:** Whether the proposed design addresses a faithfulness-specific bottleneck or is simply a general GRPO improvement.
>
> Our method was developed from analyzing faithfulness training itself. Faithfulness-critical GUI data has an unbalanced action-type distribution, with abstaining behaviors appearing as sparse reward cases (Figure 8). During training, this makes faithfulness-critical decisions more likely to form low-variance rollout groups and suffer advantage collapse (Figures 2 and 3). GuAE is introduced to address this bottleneck in faithfulness-oriented GUI-agent training.
>
> That said, we agree the underlying issue may also matter more broadly in sparse-reward agent settings, which we view as a promising direction beyond this paper.
>
> ---
>
> > **W2, Q2:** Whether the rule-based thinking–action match is effective, well-specified, and statistically justified.
>
> The consistency reward first extracts the model’s explicitly stated next-step intent from the Thought and then checks3 whether the final tool_call faithfully executes that intent. Appendix C.3 further specifies this using coarse intent categories and action-grounded cues such as typed text overlap and swipe direction, rather than unconstrained reasoning matching.
>
> The paper shows that adding $R_{Cons}$ increases the consistent portion and reduces contradictory thought–action pairs, while improving Trap performance with stable General performance (Table 2, Table 10/Appendix D.4, Table 15 / Figure 11).
>
> ---
>
> > **Q1:** How the SFT dataset is constructed.
>
> The SFT data used in Stage I is our faithfulness-oriented dataset, built by randomly sampling from the two general GUI agent benchmarks AITZ and AC. The General part uses original step-wise episodes, while the Trap part is derived from the same steps through controlled instruction perturbations and UI perturbations.
>
> We automatically annotate and normalize tool-calls, then filter invalid cases. For UI-perturbed Trap data, we additionally apply human filtering to remove low-quality perturbations.
>
> ---
>
> > **Q4:** How the method compares with curriculum-driven training.
>
> | Method | Trap Type↑ | Trap SR↑ | General Type↑ | General SR↑ |
> |---|---:|---:|---:|---:|
> | Curriculum | 90.69 | 79.47 | 81.24 | 62.79 |
> | Stage II | 91.29 | 80.21 | 85.74 | 65.62 |
>
> We compare our method with a curriculum-style baseline built from the same Stage I model and the same GRPO setup as Stage II. The training samples were partitioned into three subsets by average rollout reward and learned sequentially. As shown in the table above, the curriculum baseline remains below our full Stage II on both Trap and General. This suggests that GuAE is better suited than curriculum ordering to address the low-variance collapse that arises in faithfulness-oriented GUI-agent training.
>
> ---
>
> > **Q5:** Whether maintaining usable advantages conflicts with RL convergence toward stable action patterns.
>
> Our objective is to preserve usable learning signal for faithfulness-critical GUI decisions during training. In our setting, many of these decisions are governed by sparse, near-binary rewards, so near-zero advantages can appear before the policy has actually learned a faithful decision boundary. Figures 5 and 11 and Table 3 show that GuAE alleviates this collapse by maintaining larger non-zero advantages and smoother optimization dynamics.
>
> ---
>
> We hope the above clarifications help address your concerns and make our contributions clearer.

---

> > ### Author Rebuttal · Reviewer_oUYU · 2026-04-04
> >
> > Thank you for the responses, which addressed most of my concerns

---

### Official Review · Reviewer_GPXR · 2026-03-13

**Soundness:** 3
**Presentation:** 3
**Significance:** 3
**Originality:** 3
**Overall Recommendation:** 4
**Confidence:** 2

**Summary:**

The paper proposes Faithful-Agent, a two-stage training framework for mobile GUI agents that prioritizes "faithfulness",  grounding actions in visible screen evidence and keeping reasoning consistent with executed actions. Stage I uses SFT to teach abstention behaviors; Stage II uses GRPO with a novel Guided Advantage Estimator (GuAE) that prevents advantage collapse under sparse, near-binary rewards. Results show substantial improvement on "Trap" scenarios (perturbed evidence) while maintaining general performance.

**Compliance With Llm Reviewing Policy:**

Affirmed.

**Final Justification:**

My concerns are solved. I would maintain my original rating.

**Key Questions For Authors:**

see weakness

**Limitations:**

yes

**Strengths And Weaknesses:**

## Strengths
- The advantage collapse problem is well-motivated and the math is clean. Augmenting rollout groups with anchor points {0,1} is a simple but effective fix.
- The maximum-entropy calibration justification for $\sigma_{0} = \sqrt{(1/12)}$ is a nice theretical touch that I appreciated.
- Empirical gains are substantial, Trap SR jumping from 13.88% to 80.21% is honestly quite impresssive.
- The two failure categories (ungrounded interaction, instruction deviation) are clearly defined and easy to follow.

## Weaknesses

- The "faithfulness" dataset is self-constructed with automatic annotation, and rougly one-third of samples were discarded. Evaluation is entirely on this same distribution, which raises real generalization concerns for me.
- Trap scenarios are artificially constructed pertubations — its unclear how well this reflects real-world deployment failures. Would love to see some analysis on this.
- The thought-action consistency reward RCons is rule-based keyword matching, which seems a bit brittle for complex reasoning traces. Im not fully convinced this scales.
- Cross-dataset transfer results (Table 5) show meaningful drops, suggesting some dataset-specific tuning might be happening unintentionally.
- (Optional) The base model (Qwen3-VL-8B) is also the backbone, so comparisons against other baselines arent entirely apples-to-apples, which makes it slighly hard to isolate the actual contribution.

---

> ### Author Rebuttal · Authors · 2026-03-29
>
> We thank reviewer GPXR for the thoughtful feedback. Below we address the main points raised in this review.
>
> ---
>
> > **W1, W4:** Generalization concerns on the self-constructed faithfulness dataset and cross-dataset transfer.
>
> Our faithfulness dataset is automatically constructed from samples drawn from the widely used GUI benchmarks AITZ and AC (Table 6). Trap examples are derived from the same original steps through controlled instruction and UI perturbations. The filtered portion mainly removes invalid tool_calls or ineffective perturbations during quality control, e.g., visually implausible or ineffective masking or inpainting that fails to remove the target element, or automatically annotated actions that are invalid after post-processing.
>
> | Setting | AITZ→AC Trap Type↑ | AITZ→AC Trap SR↑ | AITZ→AC General Type↑ | AITZ→AC General SR↑ | AC→AITZ Trap Type↑ | AC→AITZ Trap SR↑ | AC→AITZ General Type↑ | AC→AITZ General SR↑ |
> |---|---:|---:|---:|---:|---:|---:|---:|---:|
> | Backbone on target set | 40.55 | 16.76 | 85.66 | 61.53 | 30.91 | 7.54 | 72.60 | 53.27 |
> | Stage I | 65.42 | 45.27 | 66.76 | 46.38 | 82.55 | 70.18 | 50.00 | 32.69 |
> | Stage II | 67.16 | 51.74 | 68.23 | 50.67 | 85.09 | 76.73 | 58.65 | 35.10 |
>
> Cross-dataset transfer shows substantial drops in General under the nontrivial distribution shift between AC and AITZ. As shown in the table above, Faithfulness-Agent with GuAE still delivers consistent stage-wise improvement from the target-set backbone to Stage I and then Stage II, especially on faithfulness-critical Trap cases. This shows that the stage-wise faithfulness gain is not confined to a single dataset.
>
> ---
>
> > **W2:** Realism of the synthetic Trap scenarios for deployment failures.
>
> Appendix Figures 12–14 provide deployment-relevant case studies showing realistic GUI failure modes, including stale visual assumptions, instruction–UI conflict, and occluded or missing targets. These reflect common information conflicts in real settings such as pop-ups and UI updates, which our Trap construction is designed to capture.
>
> ---
>
> > **W3:** Reliability and scale design of the rule-based thought–action consistency reward.
>
> $R_{Cons}$ does not semantically score full reasoning traces. It checks whether the explicitly stated next-step intent in the Thought is consistent with the final tool_call, using only a small amount of action-grounded information. This makes it a bounded alignment regularizer for thought–action consistency.
>
> The paper shows that adding $R_{Cons}$ reduces contradictory thought-action pairs and improves Trap performance while keeping General performance largely stable. The training diagnostics also remain stable, with convergent reward in diagnostics (Table 2, Table 10/Appendix D.4, Table 15 / Figure 11).
>
> ---
>
> > **W5:** Fairness of comparison when the backbone is also used as a baseline.
>
> Qwen3-VL-8B is strong overall but still weak on faithfulness-critical Trap cases, especially UI-Trap, and it is worse than specialized GUI models such as UI-TARS-1.5 on Trap. This suggests that the faithfulness gain is not simply due to starting from a stronger general backbone.
>
> Our main comparison is on faithfulness under a fixed backbone. Qwen3-VL-8B is included because it is our starting point, and the contribution of our method is isolated by the stage-wise and component ablations on the same backbone (Tables 1–3, 13). These ablations show that the gain comes from the proposed faithfulness-oriented design itself.
>
> ---
>
> We hope the above clarifications help address your concerns and make our contributions clearer.

---

> > ### Author Rebuttal · Reviewer_GPXR · 2026-04-03
> >
> > I appreciate the author for the additional analysis on the cross-dataset transfer. My concerns are solved. I would like to keep my original positive score.

---

### Decision · Program_Chairs · 2026-04-30

**Decision:**

Accept (regular)

**Comment:**

This paper studies the problem of faithfulness in a GUI agent when interacting with the environment. The problem is well presented and analyzed. The proposed methodology comprises a SFT part on abstaining behavior dataset and an improved GRPO algorithm tailored for the problem. Experiments are well conducted, which demonstrates the effectiveness of the proposed solution. All reviewers lean towards acceptance, and the rebuttal clarifies most concerns raised by the reviewers. Overall, the paper is a solid contribution towards improving the faithfulness of GUI agents. I hence recommend acceptance.